# An Anthropogenically Created Landscape as a Habitat for the European Ground Squirrel Population Using the Example of the Muránska Planina National Park in the Western Carpathians (Slovakia)

Bohuslava Gregorová [1,*], Pavel Hronček [1] and Peter Urban [2]

1 Department of Geography and Geology, Faculty of Natural Sciences, Matej Bel University in Banská Bystrica, Tajovského 40, 974 01 Banská Bystrica, Slovakia; pavel.hroncek@umb.sk
2 Department of Biology and Ecology, Faculty of Natural Sciences, Matej Bel University in Banská Bystrica, Tajovského 40, 974 01 Banská Bystrica, Slovakia; peter.urban@umb.sk
* Correspondence: bohuslava.gregorova@umb.sk

**Abstract:** The main goal of the study, based on historical-geographical research, was to reconstruct the history of an anthropogenically created cultural landscape on the southern edge of the Muránska Planina National Park in the Western Carpathians (Slovakia) as a potential European ground squirrel habitat. Humans began to change the original forest landscape in the second half of the 13th century in connection with the construction of Muráň castle, which became the economic center of the study area. The first written mention of the existence of Muráň castle (castro Muran) dates to 1271. The original deciduous forests were gradually removed and transformed into agricultural land. At the turn of the 14th and 15th centuries, we can almost certainly assume the existence of an agricultural landscape in the territory called Biele Vody (part of the study area) on the right side of the Muránsky Potok valley in such spatial dimensions as it is at present. The landscape created in this way provided suitable ecological conditions for the successful survival of the European ground squirrel (*Spermophilus citellus*). The analysis and reconstruction of the origins and development of the agricultural landscape were carried out based on detailed archival and terrain research. Map outputs are also the result of the reconstructions. Whether the European ground squirrel was already present in the locality before its conservation translocation (773 individuals were released at the site in 2000–2007) is discussed in detail in this paper. The current ground squirrel colony is dependent on feeding, mainly sunflowers (since 2011), on active management and maintenance of the landscape provided by a herd of donkeys (March–December) and sheep (May–July). The ground squirrel locality Biele Vody is currently a center of ecotourism and ecological education.

**Keywords:** agricultural landscape; historical-geographical reconstructions; European ground squirrel; landscape management; conservation; Western Carpathians



## 1. Introduction

Preserved historical land use in the current landscape reflects a similar pattern of use in the past. The preserved historical land use provides a more stable habitat reflected in its resistance to driving forces, which creates long-term suitable ecological conditions for the spread of certain (often specific) animal species. The action of dominant driving forces causes landscape changes, and when the anthropogenic process was the dominant driving force in the past, a historical cultural landscape was created. In the case of our study area, an agrarian historical landscape was formed by anthropogenic driving forces already apparent at the beginning of the modern age. Subsequently, more recent driving forces have had a less intense impact; therefore, they did not drastically alter the historical agrarian landscape. Therefore, we can consider the study area a stable cultural landscape.

Stable cultural (deforested) agrarian landscapes created in the earliest possible history represent the cultural heritage of human landscapes, thus creating suitable conditions for the survival of the European ground squirrel (*Spermophilus citellus*).

The ground squirrel is a medium-sized diurnal rodent living in colonies in open areas in the agricultural lands of Central and Southeastern Europe and Turkey. It is the westernmost representative of Palearctic ground squirrels. It began to spread in Europe roughly 5000 years ago, after Neolithic deforestation [1,2]. Its area is disjoint. The Carpathians and the Djerdap Danube Canyon divide it into two parts. The northeastern one stretches from southern Poland through the lowlands of the Czech Republic, Austria, Hungary, Slovakia, Serbia, Croatia, and western Romania. The southeastern part extends from eastern Serbia, Macedonia, and northern Greece through Bulgaria to Turkish Thrace, southern and eastern Romania, Moldova, and the Transcarpathian part of Ukraine [3–7].

The ground squirrel mainly inhabits low-grass steppes, pastures, and meadows (natural or anthropogenic), with fewer shrubs and trees, from sea level to an altitude of 2500 masl [3,8]. It occurs mainly in dry lowlands with clay-loess humus and alluvial-meadow soil types, where it mainly inhabits not only various types of mowed grasslands and pastures but also the edges of fields (especially with perennial fodder) and other anthropogenic habitats, such as lawns, playgrounds, golf courses, river dams, railways, and road embankments. In recent decades, it has developed tolerance to tall vegetation or shrubs and trees [1,9]. In the past, it was considered an agricultural pest in most of its natural range [1].

In the last half-century, its area has shrunk, and the number of individuals has significantly reduced or fragmented mainly due to the intensification of agriculture and the use of chemicals, the reduction of livestock grazing, and the abandonment of grasslands. As a result, the IUCN classified the European ground squirrel as an endangered species in 2020 [10]. Only a coordinated conservation effort at the European level can maintain its viability. Because of this, for example, the European action plan for ground squirrels was developed in 2013 [6].

The ground squirrel shows remarkable plasticity in life-history adaptation. Individuals live in the wild for three to five reproductive seasons. The maximum lifespan is four years for males and six years for females [11], but it can be more than nine years in human care.

The nominotypical subspecies *Spermophilus citellus* occurs in Slovakia [9]. In the past, it was a commonly distributed species that occurred in lowland areas and adjacent hilly areas, where it mainly inhabited pastures, meadows, terraces of fields, embankments of railway lines and roads, fallow fields, and cultural steppes, where it preferred fields with perennial fodder [12].

Due to the influence of various factors, especially the gradual abandonment of grazing by farm animals, overgrowth, or the loss of short-stemmed vegetation areas, or, on the contrary, the ploughing of grasslands, it has resulted in the fragmentation of its area and a decrease in abundance. Nowadays, it primarily inhabits grassy areas managed by humans, such as airports, pastures, golf courses, dams, and ditches by roads [6,12].

The study's primary aim is to reconstruct the (agrarian) landscape and its historical land use on the upper reaches of the Muráň stream valley in the Muránska Planina National Park (Western Carpathians, Slovakia) in Biele Vody, based on cabinet and field historical-geographical research.

The reconstruction of the historical landscape, based on analyses of maps from historical periods, allows us to document the development and stability of the study area as a suitable habitat for the life of the threatened ground squirrel population. Mapping of spatiotemporal changes is nowadays more commonly carried out in cities as part of population growth research and is related to urban planning [13,14]. The application of the historical-geographical cross-section method to research the stability of agrarian historic landscapes has not yet been implemented using this method.

Deforested agricultural land represents a potential habitat for the ground squirrel. The discussion aims to describe the parameters of the anthropogenically created landscape

and other historical driving forces from the point of view of the ground squirrel spread and to predict the time limit when the ground squirrel population could and did naturally inhabit the study area. We also intend to briefly characterize the conservation, relocation, and management of the ground squirrel colony in the anthropogenically created landscape of Biele Vody as an essential part of the population's survival in the Western Carpathians.

## 2. Study Area

The studied locality Biele Vody (with its surroundings defined as a square with a length of 2.5 km) is in the central part of the Western Carpathians on the southern edge of the highest mountains of the Carpathian arc in the middle of Slovakia. It lies at the northeastern limits of the Pannonian population expansion in Central Europe (Figure 1).

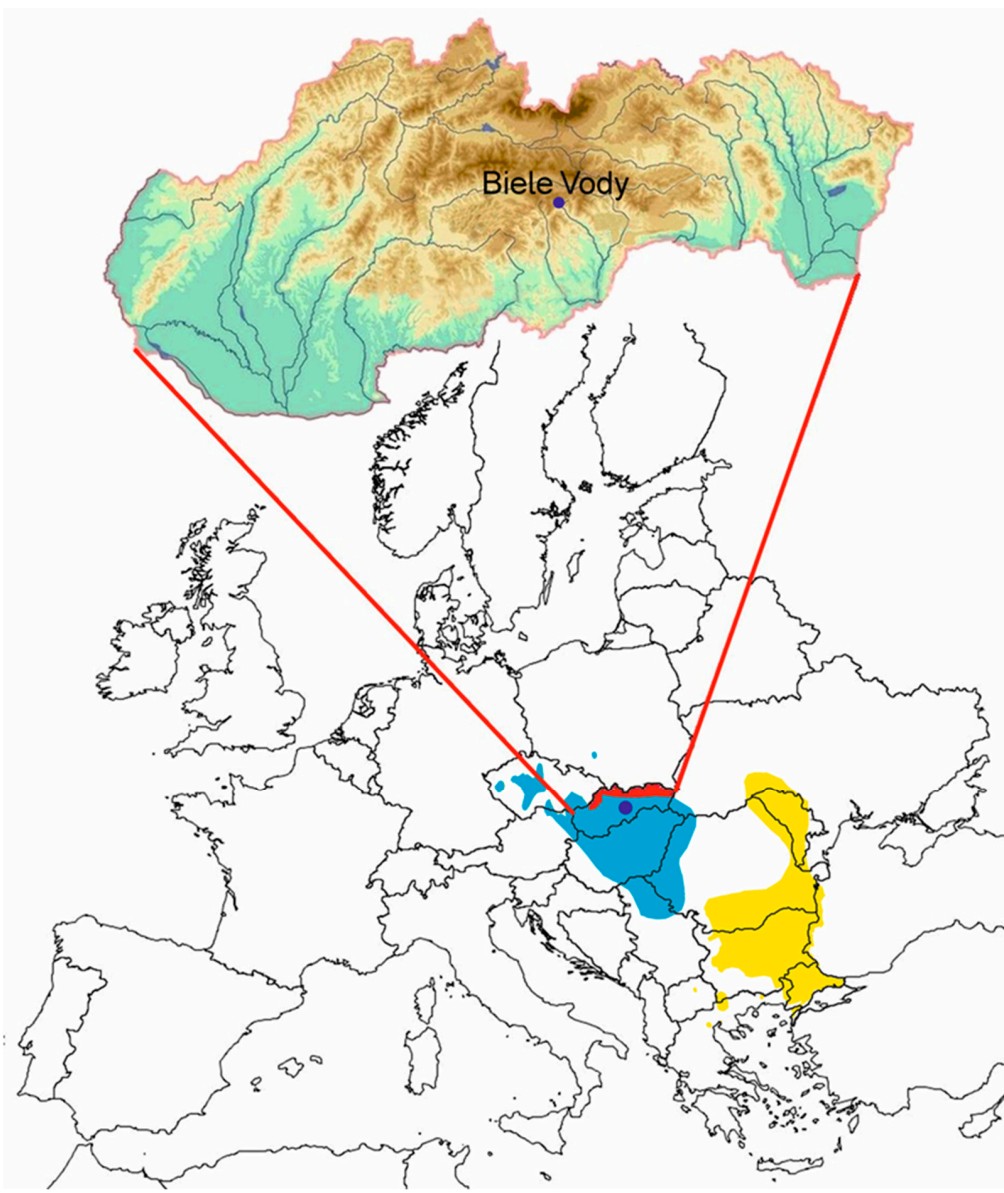

**Figure 1.** Expansion of the Balkan population (southeastern area, marked yellow) and Pannonian population (northwestern area, marked blue) of the European ground squirrel [10] and the location of the Biele Vody study area (purple point) in Slovakia (marked red). (Compiled by the authors).

It is situated in the cadastral territory of the village Muráň, approximately 2.5 km east of the residence. It is located on the upper reaches of the Muránsky stream on the southeastern edge of the Muránska Planina National Park, which was declared in 1997

(previously a protected landscape area in 1976–1996). Its current area is 18,516 ha. The national park protects a unique karst relief with original forests and minimal human intervention in the landscape. Natural ecosystems characterized by a diversity of plant and animal communities have been preserved on the territory of the national park.

The location of the Biele Vody forest site itself extends to the south-oriented, gentle right-hand slope of the Muránsky stream valley at an altitude from 430 to 455 masl. Currently, it consists of grass-herb biotopes or meadow-pasture communities. The site is farmed—conservation managed in various ways—by grazing, mowing, and mulching.

The Biele Vody locality is built with carbonate rocks, on which shallow soils—rendzins—were formed. The original oak-hornbeam deciduous forest was removed, and the locality is now a cultural steppe. The immediate proximity of the rock relief near the locality provides nesting grounds for raptors (falcons, ravens, owls, etc.), but also shelters for predators (foxes). This is an anthropogenically created agrarian locality, as indicated by the research presented in the study. The location is not threatened by communications either, as there is a line barrier of the Muránsky Potok between the Muráň−Červená Skala road [15].

## 3. Materials and Methods

### 3.1. Delimitation of the Study Area

When defining the research area, we used a square section of the historical landscape on the southeastern edge of the Muránska Planina National Park, on the right side of the valley of the upper course of the Muránsky stream. The selection of the square shape of the study area is already a proven methodology, which has been used by many authors (see Table 1).

**Table 1.** Overview of selected works in which the square shape method of the study area is used.

| Authors | Type of Research | Study Area/Country |
| --- | --- | --- |
| J. Kolejka (2002) [16] | Landscape changes of water reservoir | Nové Mlýny in the Czech Republic |
| A. Gobin, P. Campling with J. Feyen (2002) [17] | Agricultural land use aspects of the town | Ikem in southeastern Nigeria |
| T. Kuplich, C. Freitas and J. Soares (2000) [18] | Land use classification of the town | Campinas in Brazil |
| M. Boltižiar, B. Olah (2010) [19] | Historical land use changes in the biosphere reserve | Tatras in Slovakia |
| C. Munteanu, T. Kuemmerle, M. Boltižiar, J. Lieskovský, M. Mojses, D. Kaiim, E. Konkolygyuro, P. Mackovcin, D. Müller, K. Ostapowicz, V. C. Radeloff (2017) [20] | 19th-century land-use legacies and contemporary land abandonment | Carpathians/Europe |
| P. Bezák, Z. Izakovičová, L. Miklós et al. (2010) [21] | Representative landscape types of country | Slovakia |
| R. Köhler, K. Olschofsky, F. Gerard (2004) [22] | Historical land cover changes in European countries | Europe |
| P. Chrastina, J. Trojan, L. Župčán, T. Tuska, P. P. Hlásznik (2019) [23] | Land use and landscape revitalization | Slovak exclave Tardoš in Hungary |
| P. Chrastina, P. Hronček, B. Gregorová, M. Žoncová (2020) [24] | Land use changes in the historical rural landscape | Slovak exclave Čív (Piliscsév) in Hungary |
| M. Boltižiar, P. Chrastina, J. Trojan (2016) [25] | Land use development of cultural landscape | Slovak exclave Sári in Hungary |
| P. Chrastina (2008) [26] | Land use changes in the cultural landscape | Slovak exclave Nagybánhegyes in Hungary |
| P. Chrastina (2018) [27] | Land use changes in the cultural landscape | Békéscsaba in Hungary |
| P. Chrastina, M. Boltižiar (2008) [28] | Development of cultural landscape | Slovak exclave Butin in Romania |
| P. Chrastina, M. Boltižiar (2010) [29] | Land use changes in the cultural landscape | Slovak exclave Senváclav in Hungary |
| P. Chrastina, J. Trojan, L. Župčán, T. Tuska, P. P. Hlásznik (2019) [30] | Land use as a tool for landscape revitalization | Slovak exclave Tardoš in Hungary |
| P. Chrastina, K. Křováková, V. Brůna (2007) [31] | Changes in the cultural landscape | Slovak exclaves Borumlak a Varzaľ in Romania |

Source: Authors' own research.

The location and size of the square were chosen based on a detailed content analysis of historical and topographic maps, aerial and LIDAR images, and a preliminary survey of the terrain so that it covered significant landscape and relief points (elevations) within its borders. At the same time, the defined square had to have an almost ideal explanatory value concerning the topic under investigation. Here, we also relied on studying textual and historical sources and published works (bibliometric method). The subject of our research was a square of landscape with a side length of 2.5 km and an area of 6.25 km². The territory chosen in this way makes it possible to precisely define the land-use items (based on CORINE land cover classes) and simultaneously carry out their mutual comparison in particular historical periods.

*3.2. Methodology of the Research*

The research (spring 1998–autumn 2022) and the processing of its results were based on the reasonable methodological steps of scientific research: identification of the problem, the definition of the research goals, selection of research methods, creation of a research plan, data collection and analysis, data evaluation, synthesis of conclusions, and final processing of the study [32,33]. To fulfill the research objectives and process the study, we established a methodological procedure consisting of several logically connected laboratories and field historical-geographical methods [34–38]. Some methods were also used simultaneously in the research.

The first step in the identification and analysis of the anthropogenically created historical agrarian landscape of the researched territory as a potential habitat for ground squirrels was the bibliometric method combined with critical content and comparative analysis of literary and archival sources [39–41]. We simultaneously implemented a critical content and comparative analysis of historical maps [42–44]:

- Potential natural vegetation [45]
- I. military mapping from the 1780s of the 18th century; II. military mapping—1860s of the 19th century (available online: https://maps.arca-num.com/en/browse/composite/, accessed on 8 April 2023)
- aerial photographs from 1950
- digital layers of CORINE land cover from 2018 (available online: http://copernicus.sazp.sk/#mapovy-prehliadac, accessed on 8 April 2023) aimed at identifying the historical land use of the study area.

In our research, we used ArcMap 10.5 software, in which land cover maps were created based on the analysis of maps from the 1st Military mapping and 2nd Military mapping, from 1950 and 2018. We mainly focused on the percentage evaluation of changes in individual elements of the land cover (using CORINE land cover classes) and analyzed them statistically and spatially [24]. Using this method, we reconstructed the development of the historical landscape and pointed out its stability over time (see Table 2).

**Table 2.** CORINE land cover classes identified in the study area.

| Level 1 | Level 2 | Level 3 |
|---|---|---|
| 1 Artificial surfaces | 1.1 Urban fabric | Castle |
| 2 Agricultural areas | 2.1 Arable land | 2.1.1 Nonirrigated arable land |
| | 2.3 Pastures | 2.3.1 Pastures |
| | 3.1 Forests | 3.1.3 Mixed forest |
| 3 Forest and seminatural areas | 3.2 Shrub and/or herbaceous vegetation associations | Shrub |
| | | 3.2.4 Transitional woodland/shrub |

The second stage of the research was field research, which took place simultaneously with the content analysis. It aimed to identify the remnants of agrarian land use in the investigated area and verify the informative value of text, map, and image documents. During the field research, we based the methodological procedures on these works [37,46–49].

After studying the sources and familiarizing ourselves with the terrain, we applied historical-geographical analyses and syntheses using specific methods aimed at the reconstruction of the historical agrarian landscape (historical land use), primarily from the 18th century to the present, and the identification of driving forces in terms of the possible spread of the ground squirrel. We used a set of basic historical-geographical methods—the method of horizontal sections, the comparative, the retrospective, and the progressive method. When applying them, we based them not only on the methodological work of P. Hronček and B. Gregorová (2022) [38] but also on other works [37,50–52]. When reconstructing the historical landscape and predicting the spread of ground squirrels in the studied area in the past, we emphasized the importance of space and time in the transformations of the cultural (agrarian) landscape [53] when the historical landscape is perceived as a manifestation (materialization) of human activities (driving forces). Here, the method of causal analysis [54,55] is based on the cultural-geographical approach to the landscape based on the relationship between landscape—man and time [56].

In our research, we used computer modeling of historical land use using the ArcMap 10.5 software, in which we made land cover maps from the years 1782/1784, 1852, 1950, and 2018. We focused mainly on the percentage evaluation of changes in individual land cover elements and their statistical and spatial analysis. Using this method, we found out which land cover classes were changed and to which classes they were modified at the same time, as well as those that were created by man. Thus, we managed to identify the key processes that took place in the country. In this way, we were able to identify the development of grass habitats suitable for the spread of the European ground squirrel [57,58].

The last step was the final historical-geographical syntheses [49,59] and the subsequent final text processing of the study.

## 4. Results

### 4.1. Development and Emergence of the Cultural Landscape of the Studied Area (Historical-Geographical Reconstruction)

The landscape of the investigated area developed in accordance with the area of the Western Carpathians, when from the beginning of our era, the climate gradually stabilized in this geographical area, and subsequently, the geological relief of the landscape modeled by natural development was settled by the original forest-like vegetation in the younger sub-Atlantic period, i.e., after 500 AD [60]. The upper part of the Muráň stream valley, on the southern to southeastern slopes of the Muránska Planina, was no exception. An extensive primary forest formed by complexes of alluvial mountain forests, beech forests, beech-fir forests, and fir forests with fragments of heat-loving oak-hornbeam forests on south-facing slopes grew in the valley [61,62]. Primary forest, according to the definition of Prof Š. Korpeľ was (in the narrower sense of the word) an original forest unaffected by man, which in the given area, according to the species composition of the trees, conditioning changes in other features of the structure, development, and growth processes, represented the last link in the phylogenetic development of the forest. Its essential development remained permanently in the climax stage during the cyclically recurring changes of generations of diverse, long-lived woody plants [63]. These primeval forests continued to grow in the study area despite gradually intensifying anthropogenic activity up until the end of the Middle Ages. In the study area, we do not assume the existence of naturally deforested enclaves of the landscape, except for the relief areas of Cigánka and Šiance localities.

The landscape of the researched area, overgrown with impenetrable primeval forest, preserved its natural character until the 13th century when written reports first documented human activity in its wider surroundings. In 1243, in the upper part of the Muránsky stream, land properties referred to as terra Martini [64] were mentioned for the first time in written sources, with which we can most likely associate the investigated territory [65].

By analyzing the concept of *terra* in the relationship denoting the territory—the contemporary landscape in the upper part of the Muránsky stream valley at the end of the first half

of the 13th century, we can obtain some essential information for the historical-geographical reconstruction of the landscape.

The first and essential piece of information is the conclusion, according to which we can assume intensive land use at least a few decades earlier (perhaps even a century). Since, in this period, the cultivation of the land was a matter of many years (at least a whole generation), we can shift the origins of this process to the beginning of the 13th century. Therefore, it is evident that in this period, the land was used for agriculture at the expense of the original forests. We can also assume there were strips of arable land. Since the document does not mention a settlement, we can assume that the settlement was not yet completed and that a settlement in the form of a compact village did not develop here. Until the 13th century, the seasonally dispersed settlement was concentrated in the vicinity of iron ore deposits [66], primarily at the ends of the valleys on the upper reaches of the Muráň river in the wider geographical area of the studied area. The situation began to change in the second half of the 13th century because of the construction of Muráň castle, which probably occurred immediately after the Tatar invasion (1241–1242). The first written mention of the Muráň castle (castro Muran) is from 1271 [67].

The military, impregnable castle was located on the southern spurs of the Muránska Planina on the right side of the upper part of the Muráň stream. Its only defensive weakness was the saddle under the castle on the eastern side. Therefore, we assume deforestation and the creation of meadows already occurred during its construction so that it would be possible to follow the enemy's movement in the saddle. These meadows existed here until the middle of the 20th century when they were successively overgrown.

A document from 1321 [68] proves that the entire upper course of the Muráň stream was to be settled for economic use of the area, and this happened in the next few years or decades, which is documented by the first written mentions of the municipalities of the region.

The intention was to bring as many new settlers as possible to this area and create a unified and economically efficient estate. However, this process required significant deforestation and the creation of an agricultural landscape—a cultural steppe. The fact is that this process was already initiated in the first half of the 14th century, as documented by the entries in the deed, which mentions agriculturally used land, and the deed also mentions meadows in several places.

It is even assumed that in this period, there was already a grange belonging to the castle with a developed economic background in today's Muráň village [69]. We can assume that during this period, the process of deforestation, mainly of the oak-turkey forests in the valley of the upper Muráň river, was already underway, and these most favorable habitats were transformed into agricultural land. The grange most likely stood above today's village center around karst springs on the floodplain of the Muráň stream. The settlement and economic use of the upper part of the Muráň stream valley was completed during the second half of the 14th century when Muráň was mentioned as a developed agricultural village as early as 1419, which is documented from the year in question confirming the rights and privileges of the inhabitants of the village in a document stored in the archive in Budapest (Magyar Nemzeti Levéltár—Országos Levéltár, Diplomaticai Levéltár n. 24 905) [66].

Until the beginning of the 18th century, Muráň castle was the central point of the defined territory, which shaped the landscape and its use as the main transformational element. However, it burned down in 1702, and despite partial repairs from 1706, it gradually began to deteriorate [70] and became a ruin.

For the first time, the depiction of the historical landscape was preserved from the first half of the 18th century as a map of the Muráň estate from 1743 (Figure 2). The illustrated map stored in the Museum deposit in Svätý Anton provides information about the cultural landscape. The defined area's dominant feature is Muráň castle (Arx Muran), shown on the top of the Ciganka cliff as a veduta. The waterways are drawn quite accurately, not only the Muráň stream with its tributaries but also the karst spring—a hot spring under the castle and a historical road crossing the site on the right side of the valley. The map also

shows the spread of forests on both sides of the upper part of the Muráň valley, indicating their species composition. According to the Latin description *silva fagorum*, we know that the dominant tree in the forests was the beech (*Fagus sylvatica*). The intensively used agricultural landscape in the Biele Vody locality is also shown. The arable land formed by strip fields with grazed (or mowed) borders extended on the right side of the valley to the spring below the castle, confirmed by the map marker. Further on, meadows and pastures extended along the right side of the valley, confirmed by the carto-graphic mark and the Latin inscription *pratu* (meadow). The distribution of historical landscape structures indicating agricultural use and deforestation in the Biele Vody area shows only minimal changes over the last three or four centuries.

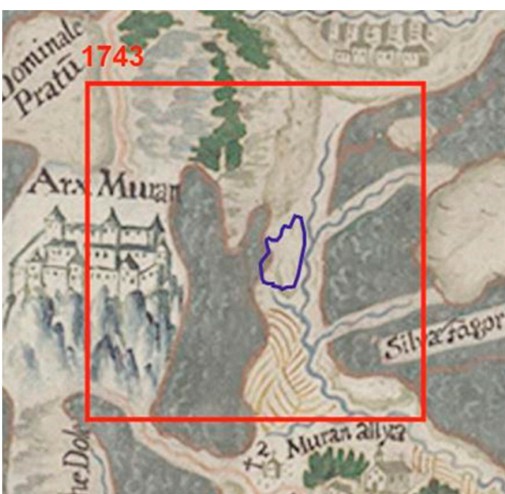

**Figure 2.** Historical land use of the study area (blue outline) is similar in extent to what was known of this area during the first half of the 18th century. Map of the Muráň estate from 1743. (Archive of the Museum in St. Anton).

A map from the end of the first half of the 18th century also documents the stability of historical landscape structures and land use of the study area with the central area of Biele Vody (Figure 3). The map from 1745 stored in the Slovak mining archive in Banská Štiavnica in the Central Chamber of Commerce fund, in the collection of maps and plans under inventory number VI-602, shows the contemporary landscape in the valley below Muráň castle (Mvran). From this cartographic document, we can deduce the essential use of the landscape in the first half of the 18th century. The valley was deforested and used for agriculture in approximately the exact boundaries as shown on the older map from 1743 and with boundaries very similar to the current borders of the land cover. The slopes of the valley of the upper course of the Muráň stream were covered with forest.

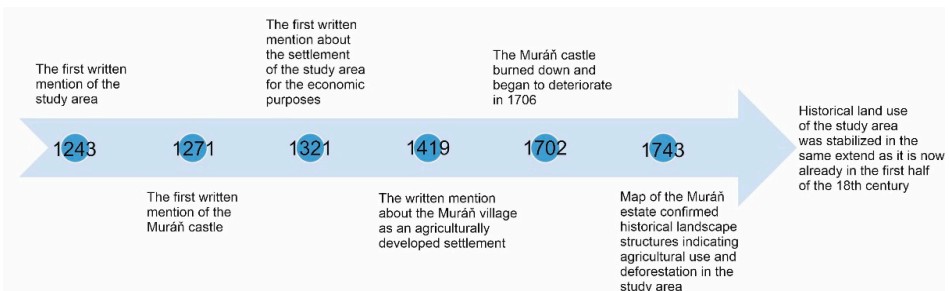

**Figure 3.** Timeline highlighting the main events influencing the development of the land use of the study area.

Based on the analysis of the land use according to the maps from the 18th century described above, when compared to the territory from the map of the first military mapping from 1782 to 1784, we conclude that the deforestation in the Muráň valley was steady for almost a century and reached a level of expansion and stability as exists currently. Based on the digitization of the map of the first military mapping (and additional information from the map from 1743), we know that the central part of the study area, i.e., today's hayfield on the right side of the Muráň stream was used as narrow strips of arable fields with borders (Figure 4), whose perimeters were bordered by meadows and pastures. Based on the digital reconstruction, the area of arable land was 38 ha, and the area of meadows and pastures was 134 ha. Arable land and pastures were mostly located at the bottom of the Muráň valley and the gentle south-facing slopes on the right. The original beech and beech-fir forest grew in the vicinity, occupying the most extensive area (442 ha). The built-up area of Muráň castle with the adjacent deforested landscape covered 6 ha.

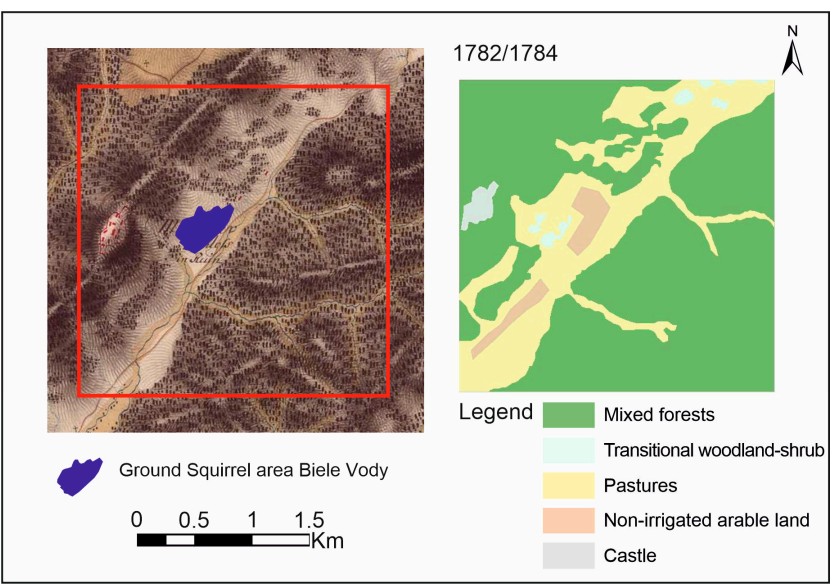

**Figure 4.** Study area from the map of the first military mapping from 1782 to 1784 and digitization of the map with defined polygons of historical land use (base map available at: https://maps.arca-num.com/en/browse/composite/, accessed on 8 April 2023).

The stability of the historical agricultural landscape in the upper part of the Muráň valley maintained by regular cultivation of arable land, mowing, and sheep and cattle pastures, is also documented by preserved drawings from the middle of the 19th century [71].

The stability of the historical agrarian landscape is also confirmed by the detailed, accurate depiction of the area of Biele Vody in the valley of the upper course of the Muráň stream seen from the map of the second military mapping from 1852. This also confirms the analysis and reconstruction of the landscape in previous periods (Figure 5). The arable land cleared as narrow strip fields, bordered by meadows and pastures, still dominated the ground squirrel territory in the study area's central part. Arable land with meadows and pastures extended along the entire right side of the Muráň stream valley, with arable land covering 45 ha and meadows and pastures covering 140 ha. The surrounding forests covered an area of 429 ha. Like the previous period, transitional woodland occupied 6 ha and the area belonging to Muráň Castle 5 ha.

The first photographs from the turn of the 20th century document the cultural forest steppe created by agricultural activity. Regarding historical land use, the upper course of the Muráň stream, including the Biele Vody site, was still used as arable land, hay meadows, and pastures. These early photos are the first real views of this landscape without the subjective perspectives of either mapmakers or artists (Figures 6 and 7). An aerial photograph from 1950 provides an accurate picture of the investigated landscape.

The aerial photograph also confirms the permanence of land use in the investigated location of Biele Vody. The floodplain of the Muráň stream was used as permanent grass areas or hay meadows. In the lower part of the slope, there were narrow strip fields of arable land along the slope's contour. In the upper part of the deforested slope, there were permanent grass areas (pastures and hay meadows). Scrub vegetation (5 ha) grew along the stony borders of this area. The upper part of the slope, up to Muráň castle, was covered with forest. The forests on the right and left sides of the valley occupied 484 ha. The land use on the right side of the Muráň stream valley, including the area of today's agricultural field, was stable, without significant changes. Arable land covered 43 ha, while meadows and pastures covered 82 ha. Transitional woodland covered 11 ha, and the location of Muráň castle was no longer visible because of the quality of the aerial photograph and the process of ongoing forest succession. A compact forest covered the castle (Figure 8). The stability of historical land use in the second half of the 20th century is also documented by topographic maps from 1955, 1964, and 1990.

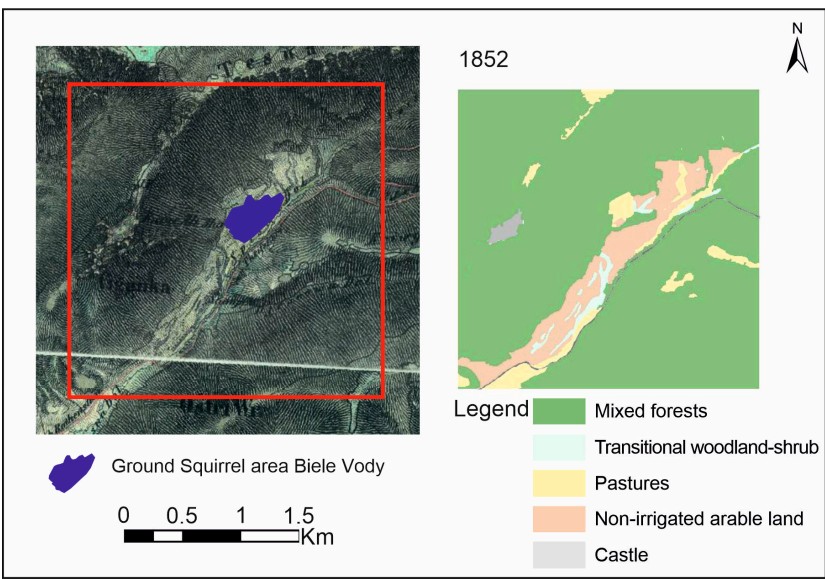

**Figure 5.** Study area on the map of the second military survey from 1852 and digitization of the map with defined polygons of historical land use (base map available at: https://maps.arcanum.com/en/browse/composite/, accessed on 8 April 2023).

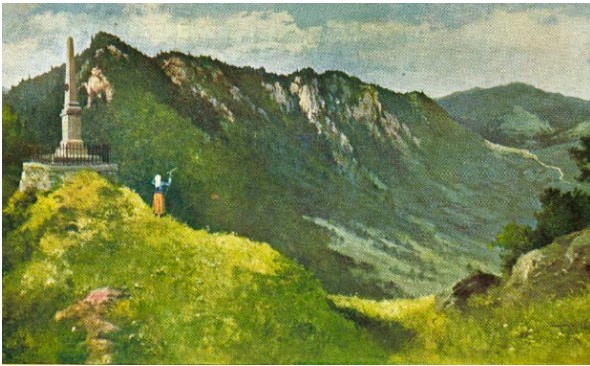

**Figure 6.** A view of the valley of the upper course of Muráň stream, which was part of the investigated area at the turn of the 19th and 20th centuries (historical postcard, authors' archive), which documents the stability of historical land use.

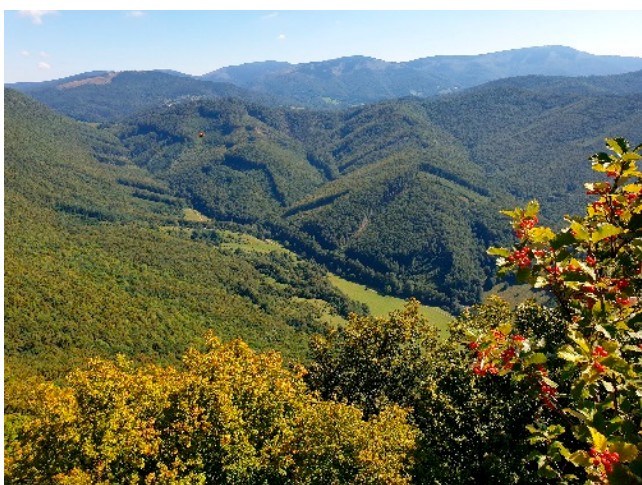

**Figure 7.** A view of the valley of the upper course of Muráň stream, which was part of the investigated area at present (photo authors), which documents the stability of historical land use.

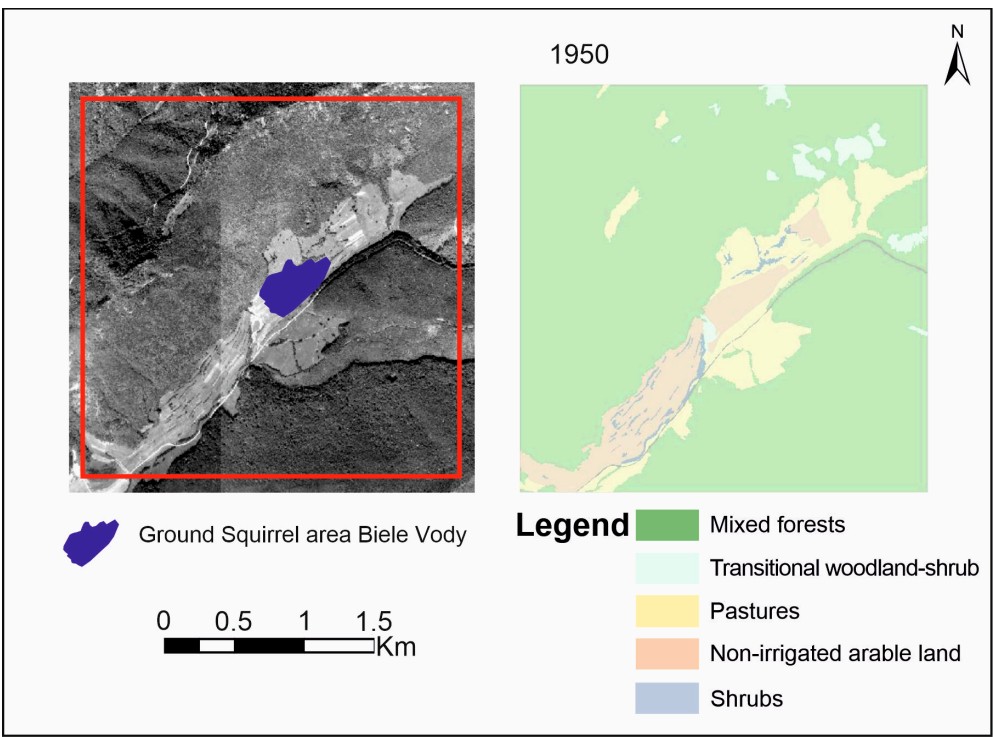

**Figure 8.** Study area on an aerial photograph from 1950 and its digitization with defined polygons of historical land use (Archive of Department of Geography and Geology, Faculty of Natural Sciences, Matej Bel University in Banská Bystrica).

The landscape of the study area and the ground squirrel field have stayed the same up to the present (Figure 9). The ground squirrel habitat is no longer arable land but has been used as meadows and pastures since the 1960s, with the total area occupying 68 ha. The forest covers 540 ha, transitional woodlands 16 ha, and the identifiable area of the castle extends to 1 ha.

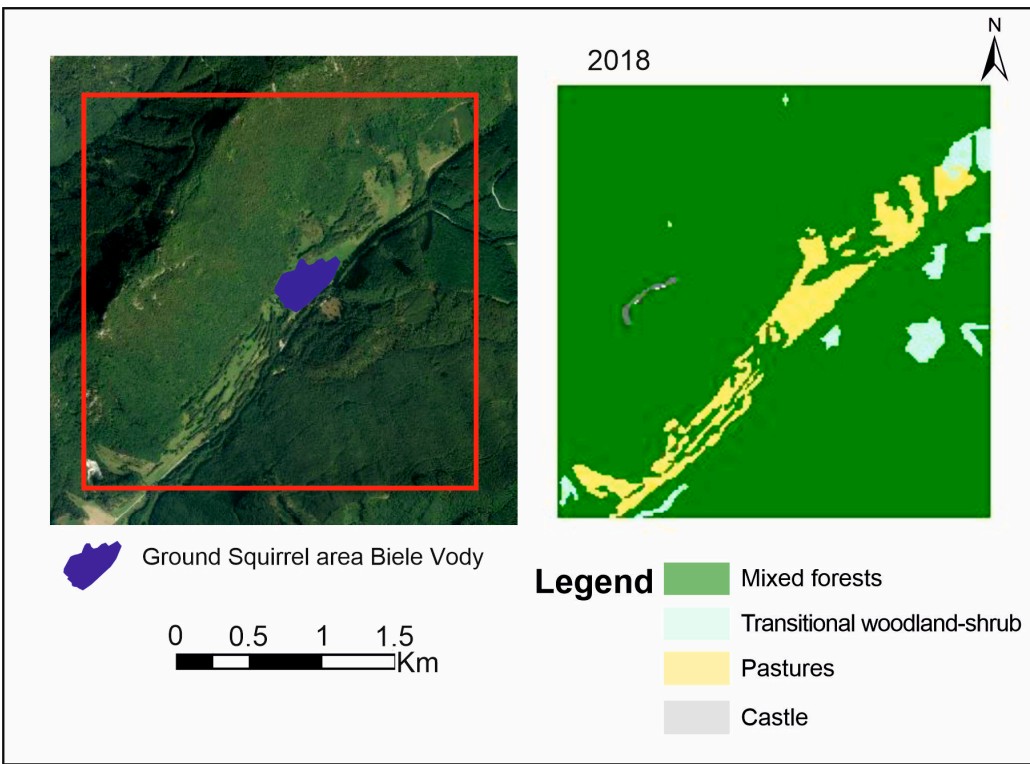

**Figure 9.** The study area in 2018 on the Google Earth image and its digitization with defined polygons of historical land use.

The historical landscape structures—as relicts of slope terraces, or rather wave-like contours across the strip fields, have been documented in addition to field research using LIDAR images (Figure 10) and Google Earth (Figure 11). From these images, we can also locate the ground squirrel burrows of the present squirrel habitat area at Biele Vody (Figure 12).

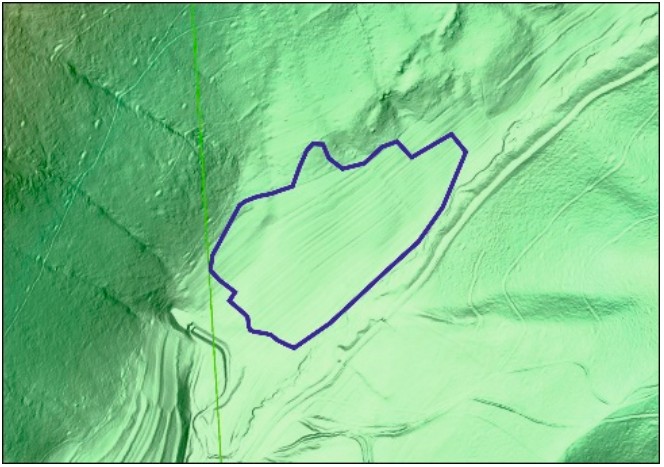

**Figure 10.** Detail of the Biele Vody ground squirrel field on a LIDAR image (source: ZBGIS).

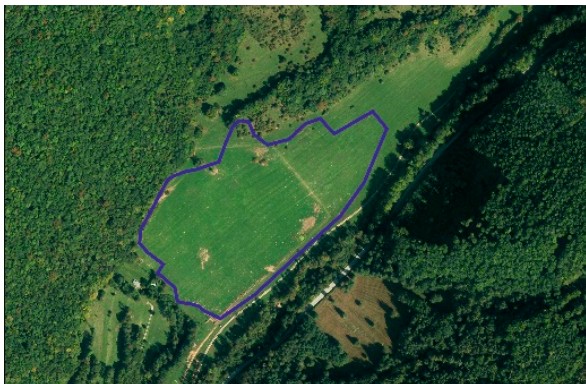

**Figure 11.** Detail of the Biele Vody ground squirrel field on an aerial Google Earth image (source https://sk.mapy.cz/zakladni?x=19.4402339&y=48.8084443&z=8, accessed on 8 April 2023).

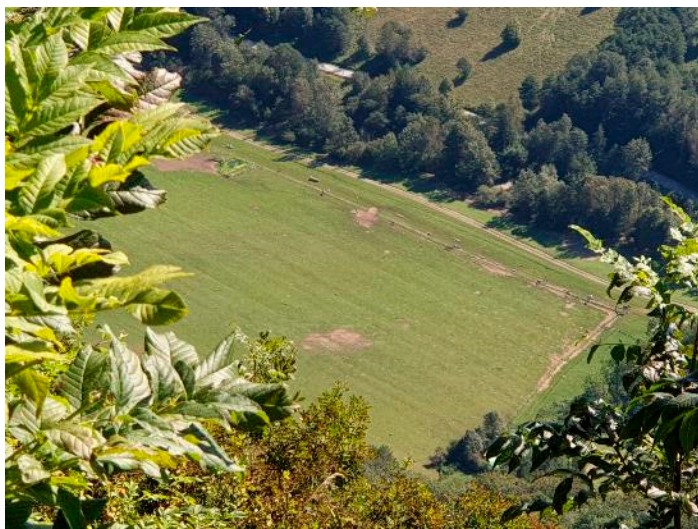

**Figure 12.** Ground squirrel area Biele Vody. View from Muráň castle (photo authors).

The historical development of land use is important for its appropriate management today. The use of historical-geographical methods, especially the documentation of land use changes over time (historical cross-section), allows us to reconstruct suitable habitats for other threatened or endemic species. Using this example of a case study of the reintroduction of the European ground squirrels, we have offered a suitable and successful research model that has also been used in the past for the study of the chamois in the Low Tatras National Park (Slovakia) [48].

### 4.2. Conservation Translocation of European Ground Squirrel in the Locality Biele Vody

The correct selection of the Biele Vody site has determined the success of the conservation relocation of ground squirrels to this locale and its subsequent successful survival and adaptation of the ground squirrel population, although these activities were and are conditioned by the application of systematic conservation management measures. The results of our research show that the location (landscape) for the survival of a small population of ground squirrels in this area was anthropogenically created already in the first half of the 18th century. Since this period, the study area has had stable historical land use. Based on our research results, the presence of ground squirrels here, despite suitable conditions in the past, even before their conservation relocation, is unlikely. Instead, this appears to be the case in the vicinity of the study area (Figure 13).

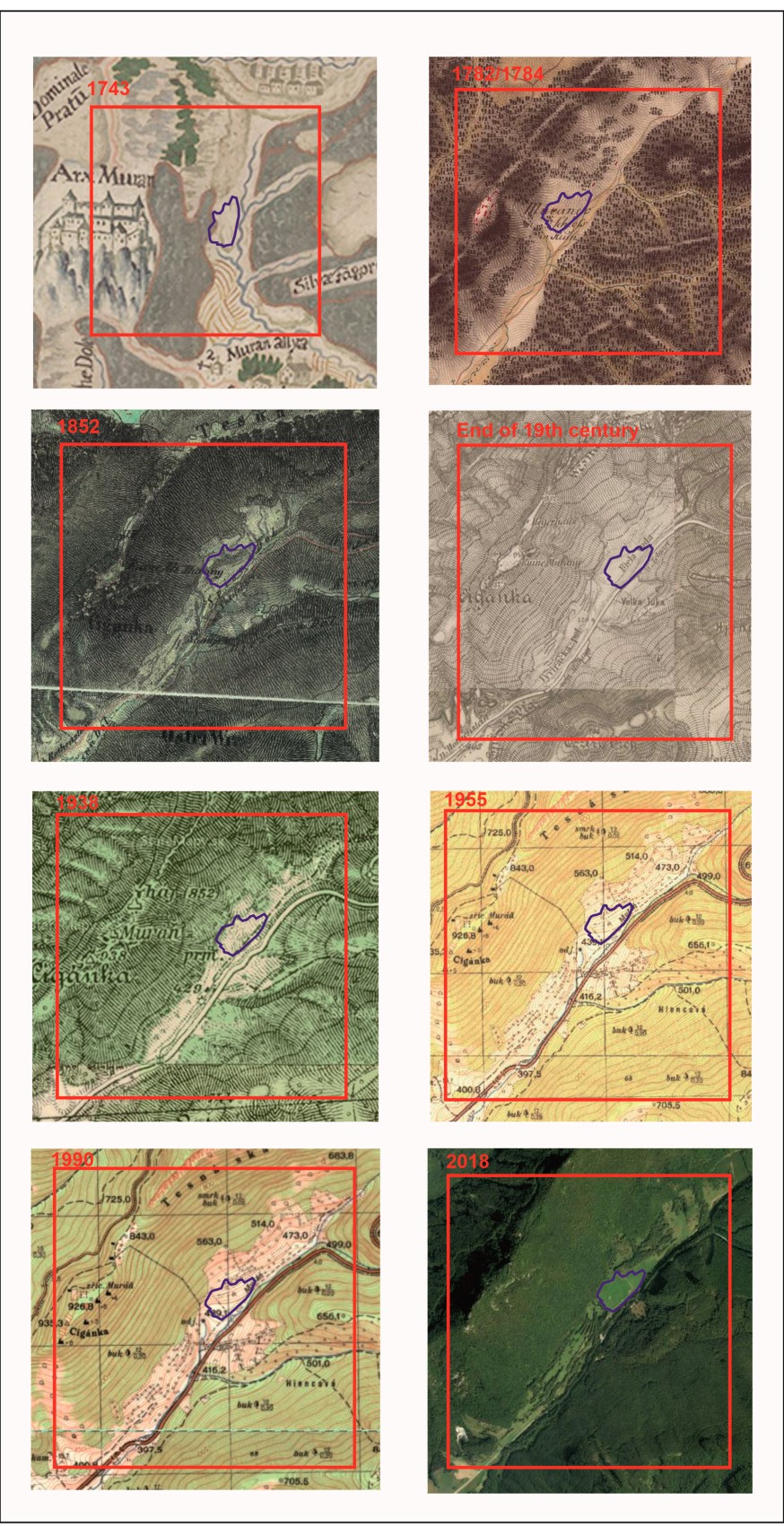

**Figure 13.** Old maps from the first half of the 18th century to the present document the stability and only minimal changes in the historical land use of the agrarian landscape of the researched area, especially the Biele Vody forest area. (Source: Museum in St. Anton, https://maps.arcanum.com/en/browse/composite/, accessed on 8 April 2023, authors' archive, Geoportal).

After selecting the Biele Vody site as a locale with a suitable and stable historical agrarian landscape in 1999, the ground squirrels were re-introduced and rehabilitated to this territory in 2000. We can define 2000−2004 as the first, less successful stage of the conservation relocation. The ground squirrels were relocated here from the airport in Košice, where they affected air traffic by attracting dangerous predators and upsetting aircraft landings and take-offs. Up to 14 ground squirrels were caught at the airport and moved to Biele Vody in these years [72]. In 2000, the first 31 individuals were released, followed by up to 161 individuals in 2001. Only 146 ground squirrels were released in 2002. In 2003, there were 153 released individuals, and in 2004, the population was supplemented by another 61 individuals [73]. At the end of the summer of 2004, the population of ground squirrels at Biele Vody was approximately 500 individuals, and the population growth continued. The following year, the agricultural cooperative in Revúca stopped grazing cows on the site, which caused it to become overgrown and spelled disaster for the ground squirrel colony. Volunteers' irregular mowing of the ground squirrel field was insufficient to create suitable conditions for a viable population. Their number began to decrease significantly despite the release of additional individuals [74].

In 2006, there were 30 of them, while in 2007, there were already 191. In 2008. up to 240 were released and increased again by 41 individuals in 2009. Altogether, 1054 individuals were released in 2000–2009 at the Biele Vody locality. Even such numerous conservation relocations did not prevent the thriving population of the 500-member colony from gradually declining to the limit of survival in 2004 when the colony had only around 50 individuals [73]. The territory was degraded and successively overgrown because of irregular mulching, mowing, or extensive short grazing. Therefore, in 2008, regular mowing of the site was initiated, which helped the population survive and stabilize.

However, in 2010, a large-scale flood affected the study area, which again decimated the ground squirrel population to its limits. In 2011, only 12 ground squirrels survived [75]. The year 2011 is the end of the transitional-unfavorable period when the stabilization and gradual increase of the population of ground squirrels began due to the change in the conservation management of the site. Since 2011, 12 donkeys have been grazing the site from March to December, and a herd of cows grazes daily from May to July. Grazing has had a very positive effect on the site; it is possible to maintain low grass communities suitable for ground squirrels, and it has also been able to stop succession. In 2011, to increase the number of the population, protective feeding began, which was carried out until 2014. In 2015, protective feeding was moved to the edge of the ground squirrel field to bring ground squirrels closer to visitors. In 2018, it was switched to direct feeding by visitors, aiming to increase the attractiveness and visitation of the site, education, and financing of ground squirrel protection. These management measures did not negatively affect the population, but on the contrary, they had a very positive effect. The population grew to about 4000 individuals in 2019 (see Figures 11 and 12 for details).

The following year, however, it was greatly devastated by a catastrophic flood, and the number of individuals fell below 1000. Nowadays, the population is growing again and prospering. In 2017, all the ground squirrels at the Biely Vody site were fed. In 2017, the ground squirrel population was distributed on the mother ground of Biele Vody (on the right side of the Muráň river valley) in a length of approx. 480 m, and in 2018, the length of the inhabited area already reached 3 km (Figure 14). However, independently living enclaves are not fed [76]. See figures in Appendix A (photos by authors).

In addition, other management activities (e.g., planting of fruit trees) are taking place on the site and its surroundings, the aim of which is to increase the biodiversity of the area and to help the spontaneous return of other species of organisms that were present at the site but were lost due to a change in its use.

There were several reasons for the increase in the ground squirrel population. The first is the protection of ground squirrels to create a foodbase for predators. Ground squirrel predators have been confirmed on the ground squirrel field, e.g., *Corvus corax*, *Accipiter gentilis*, and *Felis silvestris*. The second is creating a suitable genepool base for further

conservation relocations to other locations (in the vicinity, e.g., under Okrúhla skala) and, finally, the popularization of the issue among the general public.

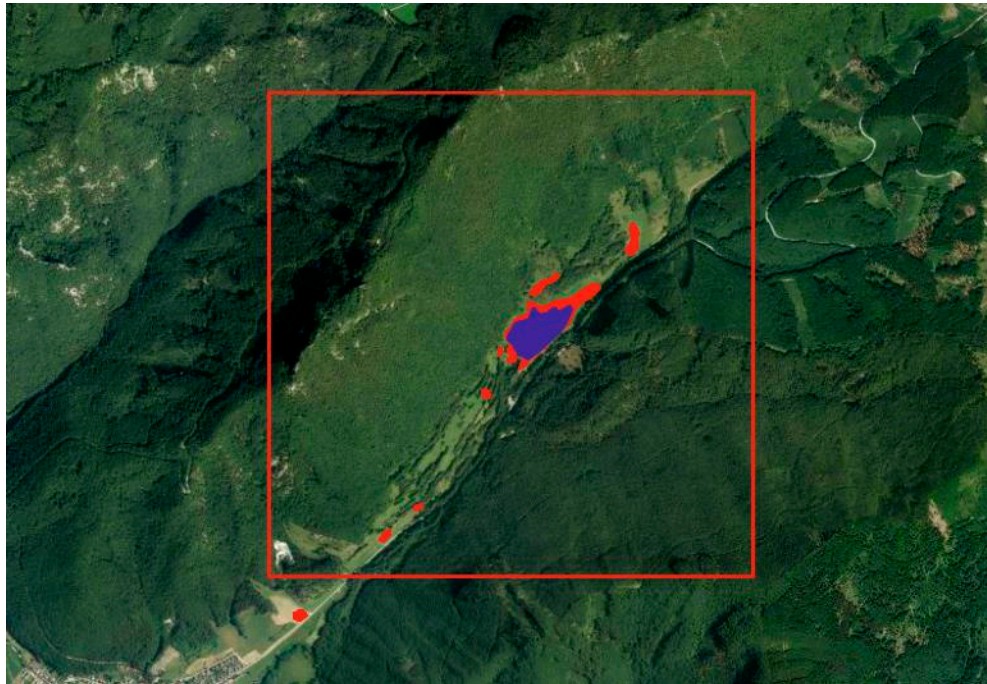

**Figure 14.** The location of the mother ground squirrel field at Biele Vody (blue area) until 2017 and the creation of other inhabited localities by ground squirrels' natural migration (red areas) [76].

## 5. Discussion

A comparison of the land-use stability in the defined area, but especially in the current Biele Vody ground squirrel field, in the course of history, based on computer analyses of land cover layers, points out that the landscape in the Biele Vody area showed almost unchanged land-use elements since the 18th century. We obtained basic information about the boundaries and area of individual land uses from the map of the first military mapping, which we supplemented with historical documents, other historical maps, and image sources. We can state that the anthropogenically created deforested agricultural landscape in today's ground squirrel habitat already existed at the beginning of the 18th century. In the study area, the gently sloping southern slopes were used as arable land in strip fields with adjacent meadows and pastures on the steeper slope. This cultural landscape character remained the same even in the second half of the 18th century. Diminutive changes occurred when scrub vegetation gradually took hold on the field borders or agrarian stone clearings were created by collecting stones and documented by contemporary rock piles. Research has proven that the ground squirrel's success depended on unchanged agrarian use, and thus the entire landscape until the middle of the 20th century (Figure 8). The change in historical land use, favoring the natural spread of ground squirrels, occurred only at the end of the 1950s, after collectivization (nationalization of land) and the creation of common agricultural cooperatives. The area of Biele Vody ceased to be used as arable land and began to be used as meadows and pastures throughout the entire area, as is the case even today (see Table 3). Little significant wavelike contours along the narrow strip fields have been preserved until the present day, which can be verified not only by field research but also by LIDAR images (Figure 10).

**Table 3.** Development of land use in the study area during historical periods (in hectares).

| Type of Map | Castle | Nonirrigated Arable Land | Pastures | Mixed Forest | Shrub | Transitional Woodland-Shrub |
|---|---|---|---|---|---|---|
| I. Military mapping from the 1780s of the 18th century | 6 | 38 | 134 | 442 | 0 | 6 |
| II. military mapping from the 1860s of the 19th century | 5 | 45 | 140 | 429 | 0 | 6 |
| Aerial photograph from 1950 | 0 | 43 | 82 | 484 | 5 | 11 |
| CORINE land cover from 2018 | 1 | 0 | 68 | 540 | 0 | 16 |

Source: Author's own research.

The stabilized historical agrarian landscape created a good environment for expanding the ground squirrel population in the Biele Vody area in the 18th century. Nevertheless, other essential factors also affect its natural expansion. Therefore, to be able to fulfill part of our original research goal and answer the question of when, or if at all, the ground squirrel population could naturally inhabit the studied territory, we must also critically assess the other results obtained during the laboratory and field research. In particular, the parameters of the anthropogenically created landscape and the historical driving forces (natural and anthropogenic) that prevailed in the studied area. Our research almost completely ruled out the presence of not only a larger population but also individuals of ground squirrels in the study area until the middle of the 13th century because the study area was dominated by a primeval forest landscape. All economic activities up to the first decades of the 14th century did not have a fundamental impact on the transformation of the landscape and the creation of larger deforested (agricultural) areas, which would have created suitable habitats for the expansion of ground squirrel colonies on the upper reaches of the Muráň stream, including the studied area. The situation changed very slowly in the third decade of the 14th century. However, the landscape did not yet have enough vast areas of deforested and agricultural land to meet the ecological requirements (spatial and food) for the spread and survival of this species.

Based on the results of our research, already at the turn of the 14th and 15th centuries, we can almost certainly assume the existence of a cultural (deforested) agricultural landscapes even in the investigated territory of Biele Vody in the exact spatial dimensions as it is today. The spatial expansion of arable land and meadows in this area is documented by old relics of agricultural terraces and, above all, the expanse of Muráň castle territory. The oldest depiction of the castle by Augustín Hirschvogel from 1549 already documents a stabilized cultural landscape in this area [77]. The original depiction of the castle is stored in the graphic collection of The Albertina Museum Vienna under inv. no. DG 1930/2163. The intensively used agricultural land and deforested landscape on the upper course of the Muráň stream is also documented in the second half of the 16th century by the surveyor of the Muráň estate from 1573 [78]. From the point of view of the existence of a cultural steppe—a deforested agricultural landscape, already in this period, i.e., at the beginning of the modern era, sufficient landscape space existed for ground squirrels to survive. Here, however, the factor of abundance, or lack of food, significantly limits their population expansion. According to the land register of the Muráň castle domain from 1573, basic cereals (wheat, oats), then legumes and fodder were grown in the fields, and meadows and pastures were also intensively mowed and grazed.

Nevertheless, a big problem was the abundance or lack of crops as food since it was in this period (half of the 16th century) that the Western Carpathian territory entered the period of the so-called Little Ice Age (hereafter LIA) [79]. The LIA period began to

manifest itself in the Carpathian region with frigid years after 1560 [80] and lasted until the 1890s [81]. The entire LIA period is characterized by unfavorable and often extreme weather in individual years and during the summer and winter half-years. In Upper Hungary, the LIA was manifested (compared to the long-term characteristics of the weather in the 20th century) by colder and wetter weather with prolonged and colder winters. The snow cover often lasted until the spring months or the middle of the spring season. It was typical for cold winters that they followed each other in more extended series, significantly worsening the situation for the inhabitants and the animals. The unfavorable winter half-year was often followed by highly unfavorable weather in spring and summer. An almost typical phenomenon was the late start of the growing season, which gave agricultural crops little time to ripen. Therefore, the stands were weaker and less resistant not only to the weather but also to diseases. Fungal diseases regularly attacked cultivated crops. Another factor was the high humidity caused by the large rainfall volumes. Numerous, often extreme storms were accompanied by strong winds, downpours, and hail. Glacial hailstones reached such a size that they destroyed the crops in the fields, threatened people and their farm buildings, and many times killed livestock. Torrential rains caused devastating floods in watercourses, and the upper course of the Muráň stream was no exception due to extensive deforestation. Wet and cold summer half-years were replaced irregularly by extremely long-lasting heat and drought, which dried up the land and destroyed food crops and pastures, causing livestock to suffer and often perish (including wild animals). The parched landscape and wooden buildings covered with straw or shingles were threatened by fires during the dry season. These extreme weather fluctuations and natural disasters were followed by famines, which often became epidemics [82].

Even though historical maps from the first half of the 18th century document a stable agrarian landscape in the investigated area of the upper course of the Muráň stream (Figure 13), we do not expect ground squirrel colonies to expand in this area, primarily due to the lack of food. Unfortunately, no written documents have been preserved that would confirm or refute the natural distribution of the ground squirrel in this geographical area. Nevertheless, we do not entirely rule out the presence of ground squirrels in the form of several individuals or families, especially in the second half of the 19th century. In that period, the influence of the LIA gradually disappeared, and the weather stabilized, which impacted the increase in the harvest. Agricultural yields also increased due to improved agricultural practices, which increased the harvest in the Carpathian region. Our assumption can also be supported by research from the first half of the 20th century, which describes the occurrence of individuals or small colonies of ground squirrels in the foothills of agricultural areas between sloping narrow strip fields separated by borders. The field borders were grazed by farm animals or mowed and represented suitable sites for the establishment of burrows, while the adjacent fields provided food. However, the ground squirrel was considered a pest and was intensively hunted and killed. These researches also confirm that local inhabitants remembered such sightings of ground squirrels several decades earlier. The research of E. Kyseľová [83] also confirms the existence of ground squirrels as several individuals or small colonies on the upper reaches of the Muráň stream at least as early as the beginning of the 20th century. A small colony was supposed to survive on the left slope of the Muráň valley near the road between Muráň and Muránska Huta, which would be only a few hundred meters from the current Biele Vody ground squirrel field. At the current state of research, we cannot say with certainty when this ground squirrel field disappeared. Based on the general factors that influenced ground squirrels in the Western Carpathians, we can assume that it happened during the collectivization period at the end of the 1950s or even (which is less likely) during the intensive use of chemical products for agriculture in the 1980s. Reports from residents who remembered the occurrence of individuals in the 1950s and 1960s [10] could also point to a certain continuity of the ground squirrel settlement in the vicinity of Biele Vody.

The ambiguous, rather improbable presence of ground squirrels in the upper reaches of the Muráň stream in the 1970s is brought to our attention by the research of J. Obuch,

who carried out a systematic collection and evaluation of the osteological material of the predator's droppings in Muránska Planina National Park between 1978 and 1983 [84,85]. A relatively high share of ground squirrels was recorded in the food of the peregrine falcon (Falco peregrinus) and the jerfalcon (*Falco cherrug*), which, however, have a long-range hunting radius. V. Hanák and M. Anděr [86] also question the presence of ground squirrels on the Muránska Planina during this period because they did not record its presence during the research of small mammals on the plain.

Thus, our research confirmed the ground squirrel's presence in the upper reaches of the Muráň stream valley only after its conservation relocation to the Biele Vody site in 2000.

The research was carried out because of the need for practice for the Muránska Planina National Park and CA Živá Planina. Several catastrophic natural events (floods) have decimated the ground squirrel population in the study area, and conservation organizations needed to know the history of the establishment and economic use of the locality to set up its proper management and the management of the ground squirrel population itself. The research clearly confirmed that this is an anthropogenically created locality which, without further management, would be heading towards degradation (due to succession) and the disappearance of the ecological conditions for ground squirrel life.

## 6. Conclusions

Our laboratory and field historical-geographical research unequivocally point out that the agrarian landscape in the foothills and mountains of the Western Carpathians (Slovakia) began to emerge at the end of the Middle Ages when the settlement of these areas was definitively completed. The research confirmed that the typical agrarian landscape with low-grass formations of meadows or pastures and arable land arranged in narrow strip fields with grass borders was stable at least by the first half of the 18th century. During the preliminary survey of historical documents, this was the case at the Biele Vody site and the entire Muránska Planina National Park area. A preliminary critical historical analysis of these documents points to a stable cultural agrarian landscape suitable as a habitat for the ground squirrel already in this period. Such a landscape was in the space of the other currently existing ground squirrel habitat on the southwestern, northern, and northeastern edges of the Muránska Planina National Park. In the future, however, this preliminary survey must be supported by systematic and complex historical-geographic archival and field research.

Although we have proven the presence of a stable historical agrarian landscape since the 18th century, its use of habitat by the ground squirrel has still not been completely confirmed. The main opposing argument is the unfavorable weather during the LIA period, the extreme conditions further multiplied by the relief (i.e., mountain ridges and deep valleys) in the Western Carpathians. This unfavorable situation was reflected in the lack of food for many years. Since the extreme course of LIA weather in the Western Carpathians ended in the second half of the 19th century, it is possible to assume a gradual settlement. Our research indicates that a stable historical agrarian landscape, or its remnants in the present landscape, are suitable habitats for the conservation relocation of ground squirrels. The conservationists themselves admit that the first and most crucial step for the survival of displaced individuals is the appropriate selection of a site.

Our research confirms that a comprehensive and systematic archival and field historical-geographical analysis of selected locations is suitable for identifying such territories. The researched Biele Vody ground squirrel location was appropriately selected, and the correct conservation management procedures were subsequently established. The management of ground squirrel relocation in this way met the conservation requirements for preserving the threatened ground squirrel species in the Western Carpathians and created a genepool from which the ground squirrel could be moved to new locations for conservation purposes. It also created a good food base for predators, whose protection is essential in the national (Slovak) and the European predator protection scheme. Finally, this prosperous ground squirrel field becomes a critical educational space and a stimulus for the development of

ecotourism as a type of nature tourism (low-impact nature tourism), contributing to the protection and care of rare species and their habitats [87], within the Muránska Planina National Park.

Management and sustainability of the ground squirrel field are provided by the CA ŽiváPlanina. The management of this locality is ensured in accordance with the laws of the Slovak Republic. CA members maintain the locality against succession by mowing and controlled burning. Active maintenance of the locality is also ensured by the permanent presence of donkeys and grazing by flocks of sheep in the morning and evening hours. Also, supplementary feeding is ensured not only by CA and national park workers but also by visitors after education programs about the ground squirrel are introduced. For larger groups of tourists or school trips, CA workers also provide eco-education directly on the ground squirrel field. CA Živá Planina also promotes the ground squirrel field on the internet with its own website https://www.syslovisko.sk/ (accessed on 23 June 2023).

**Author Contributions:** Conceptualization, P.H. and P.U.; methodology, P.H.; software, B.G.; validation, P.H., P.U. and B.G.; formal analysis, B.G.; investigation, P.H., P.U. and B.G.; resources, B.G.; data curation, P.H., P.U. and B.G.; writing—original draft preparation, P.H.; writing—review and editing, P.H., P.U. and B.G.; visualization, B.G.; supervision, P.H.; project administration, B.G.; funding acquisition, P.H., P.U. and B.G. All authors have read and agreed to the published version of the manuscript.

**Funding:** This research and the APC was funded by Slovak Research and Development Agency, grant number APVV-18-0185 "Transformácia využívania kultúrnej krajiny Slovenska a predikcia jej ďalšieho vývoja".

**Data Availability Statement:** The data used to support the findings of this study can be made available by the corresponding author upon request.

**Conflicts of Interest:** The authors declare no conflict of interest. The funders had no role in the design of the study; in the collection, analyses, or interpretation of data; in the writing of the manuscript, or in the decision to publish the results.

## Appendix A

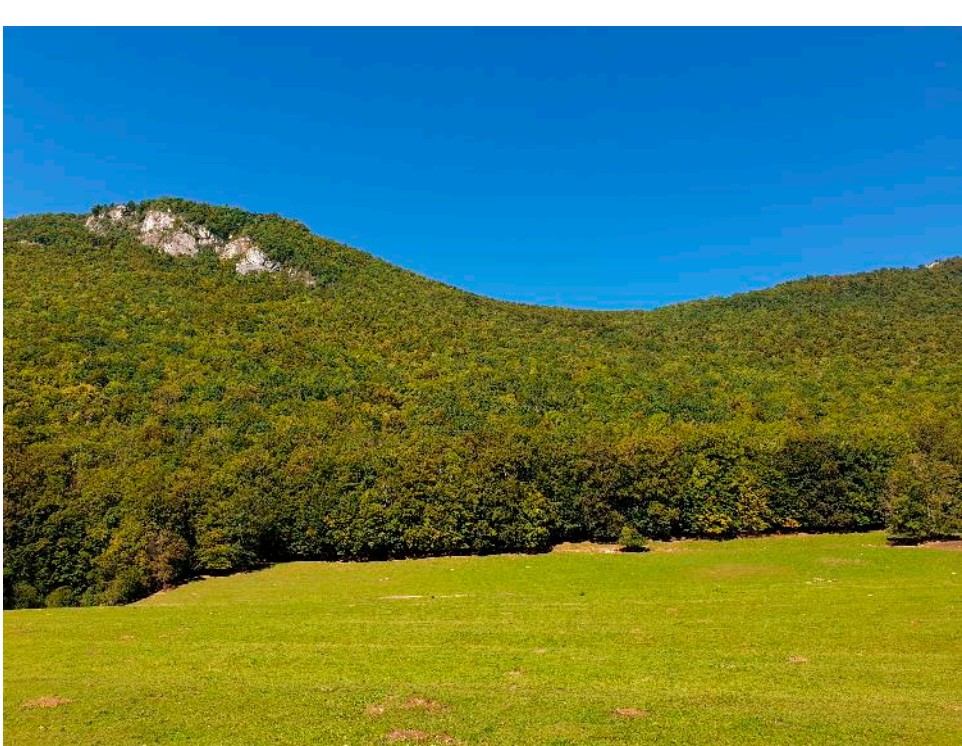

Ground squirrel field Biele Vody.

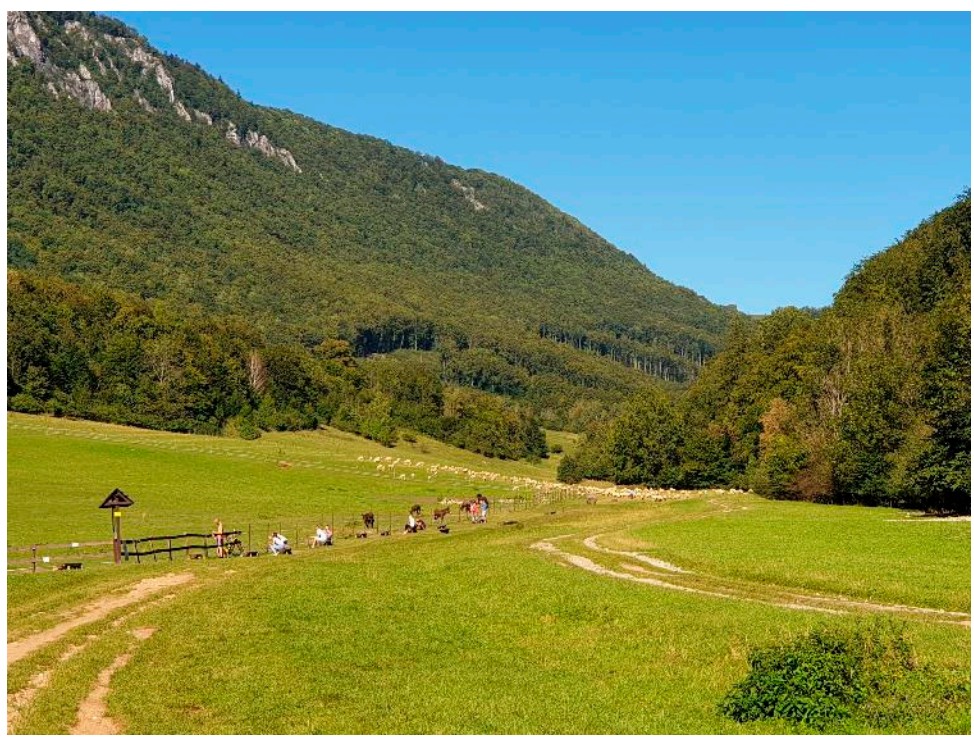

Tourists near the ground squirrel field Biele Vody.

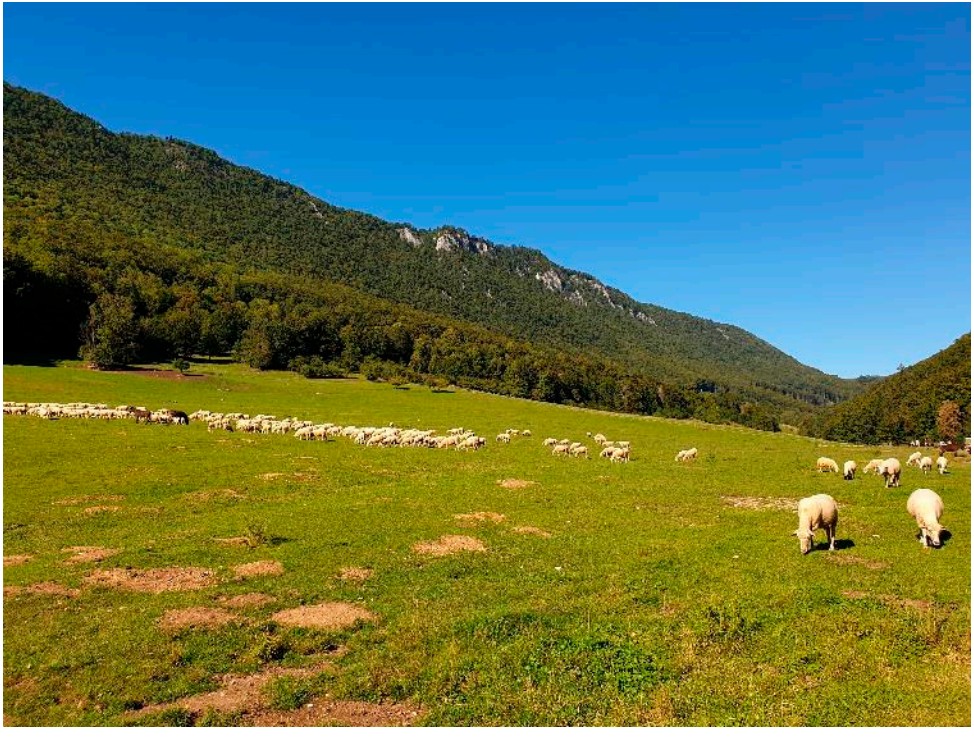

Sheep grazing in the ground squirrel field.

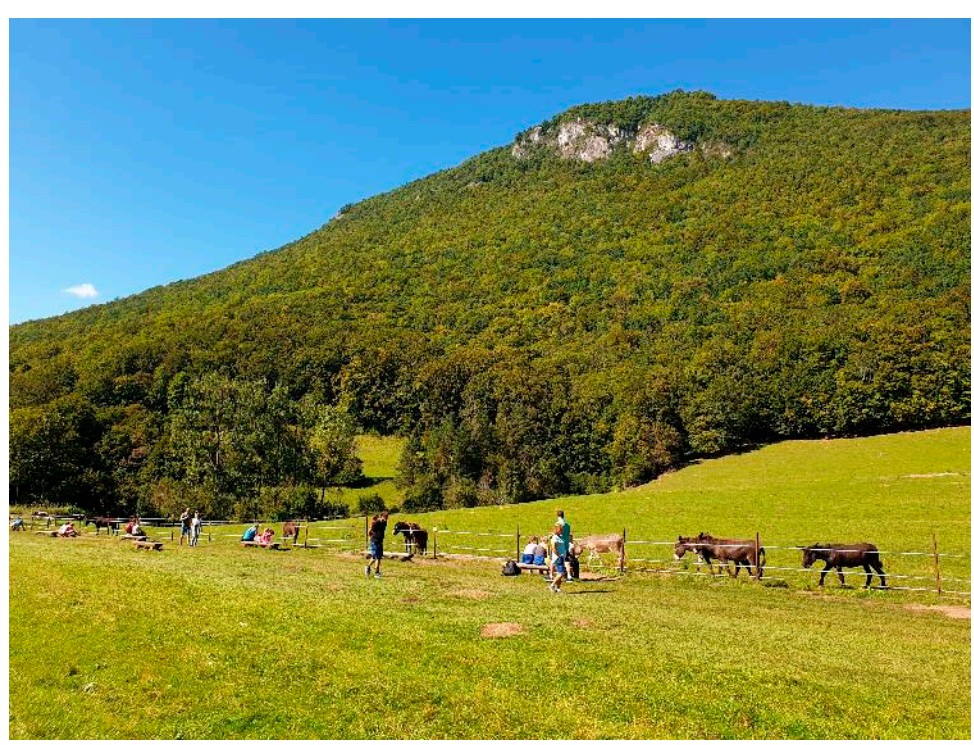

Grazing of donkeys in the ground squirrel field.

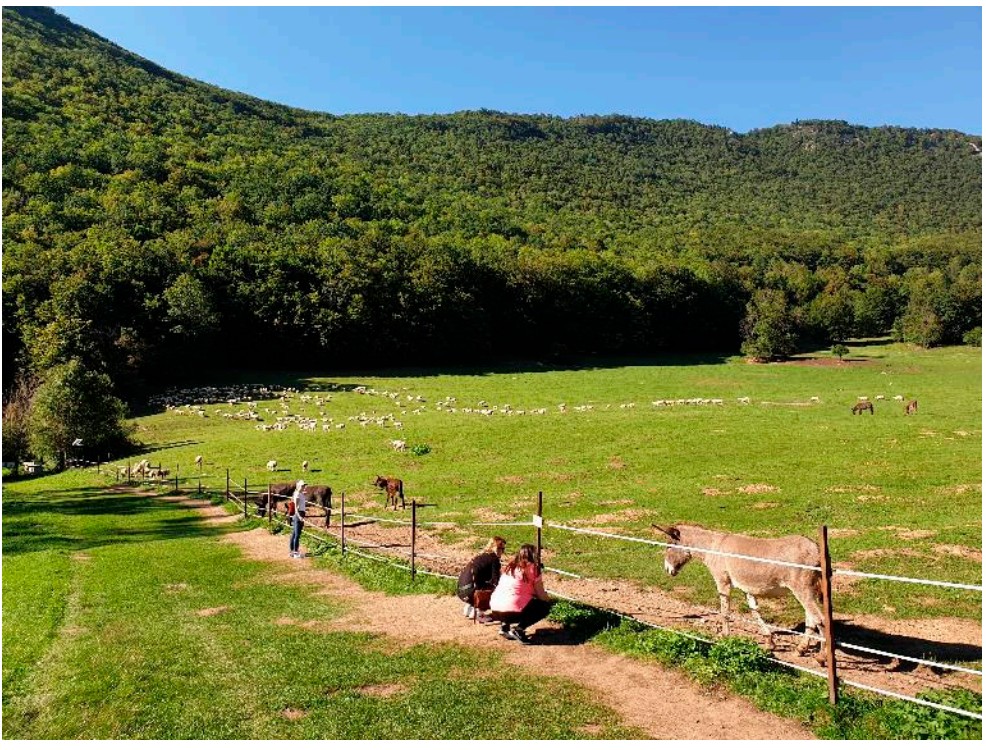

Grazing of sheeps and donkeys in the ground squirrel field.

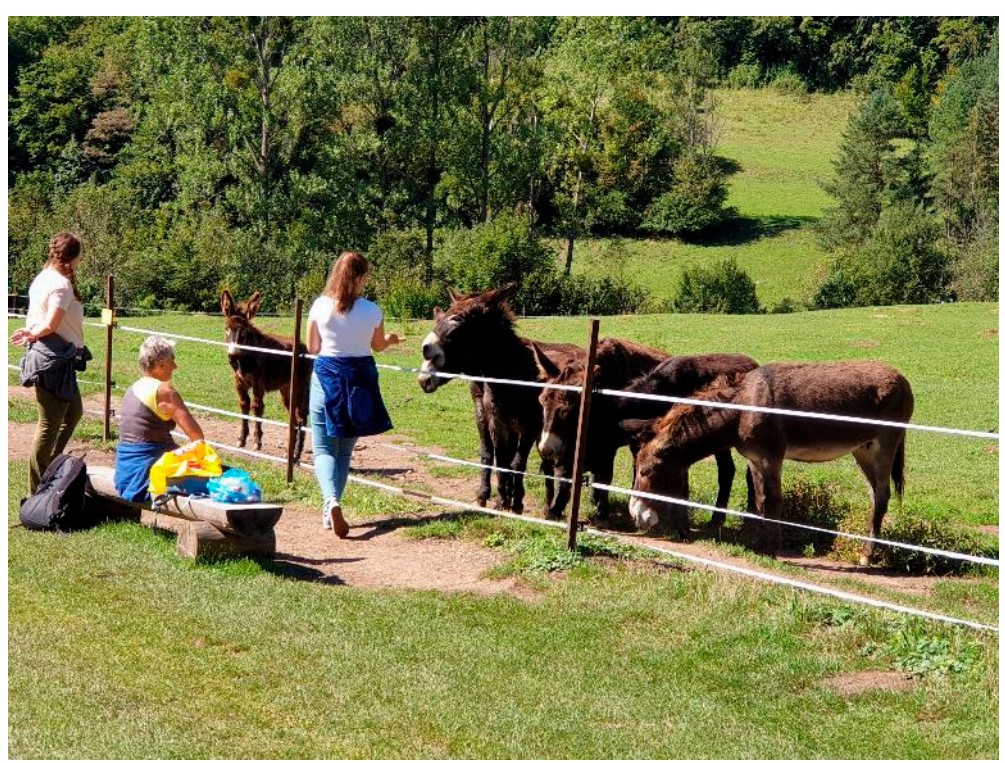

Feeding donkeys by tourists.

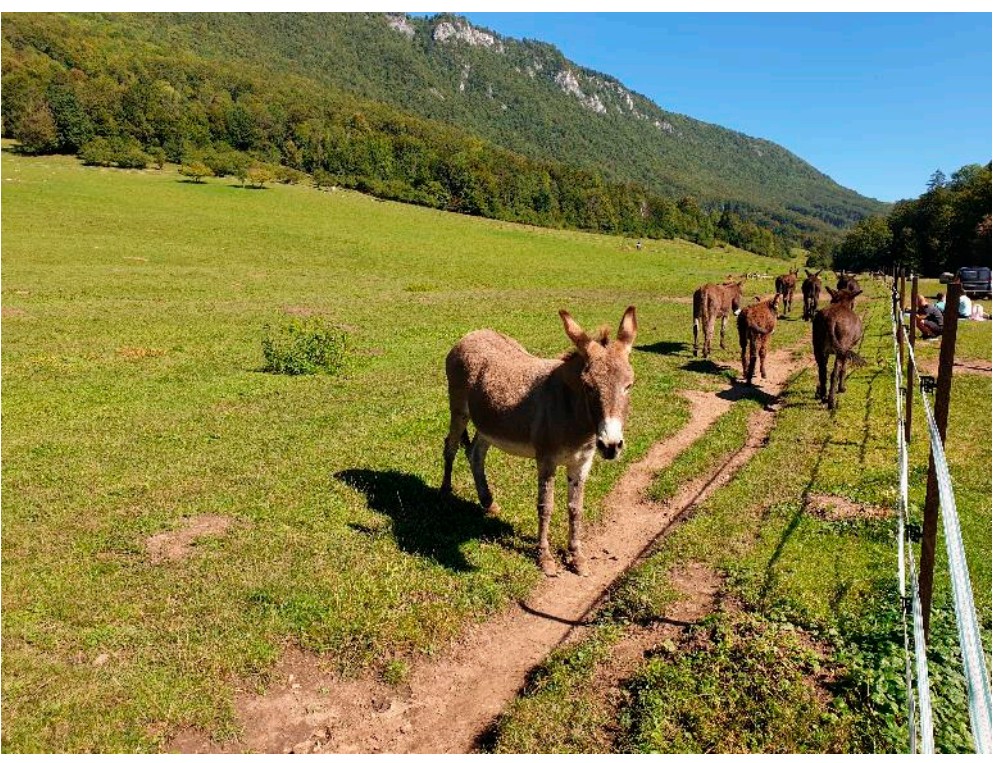

Donkeys in the ground squirrel field.

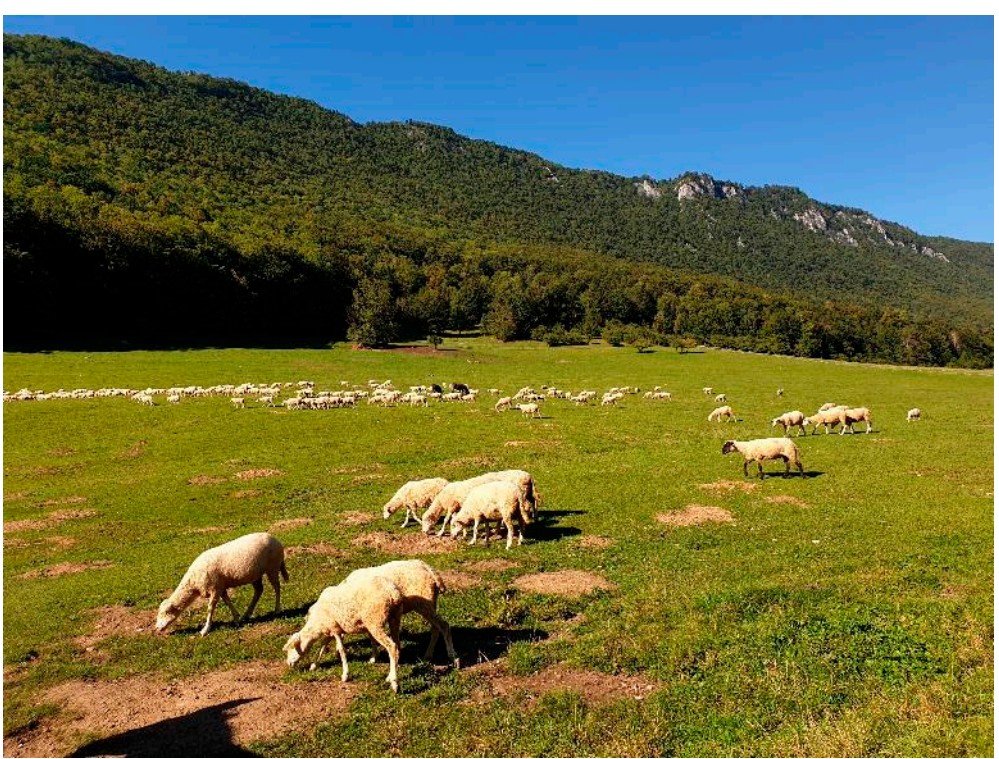

A herd of sheeps in the ground squirrel field.

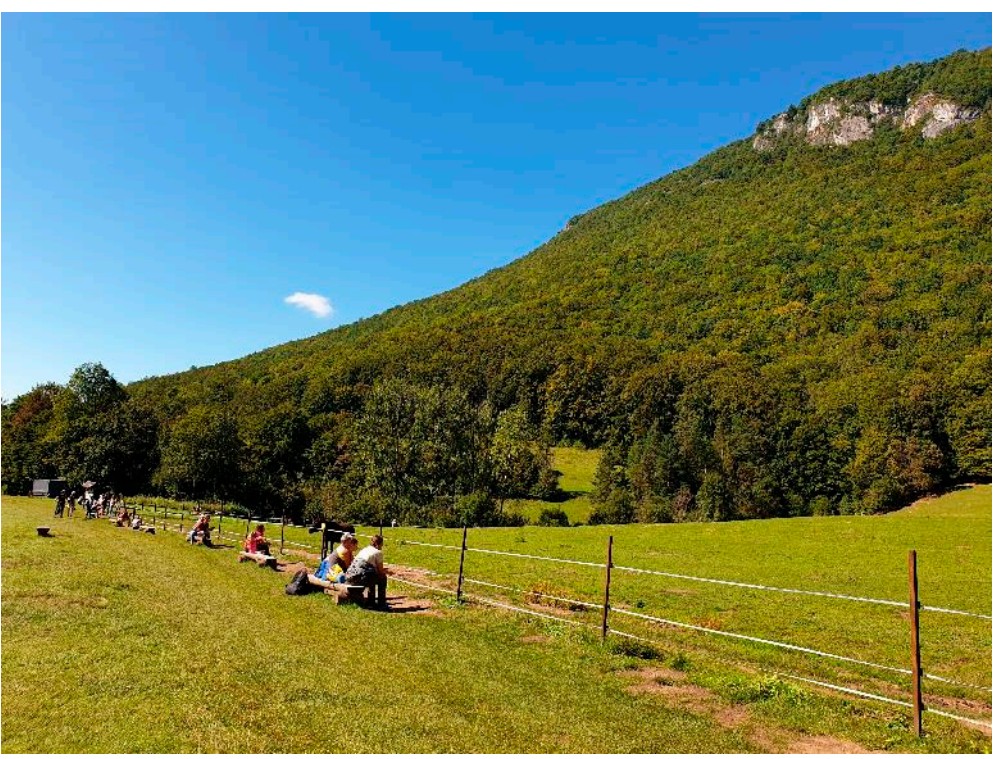

Tourists feeding the European ground squirrel in the ground squirrel field.

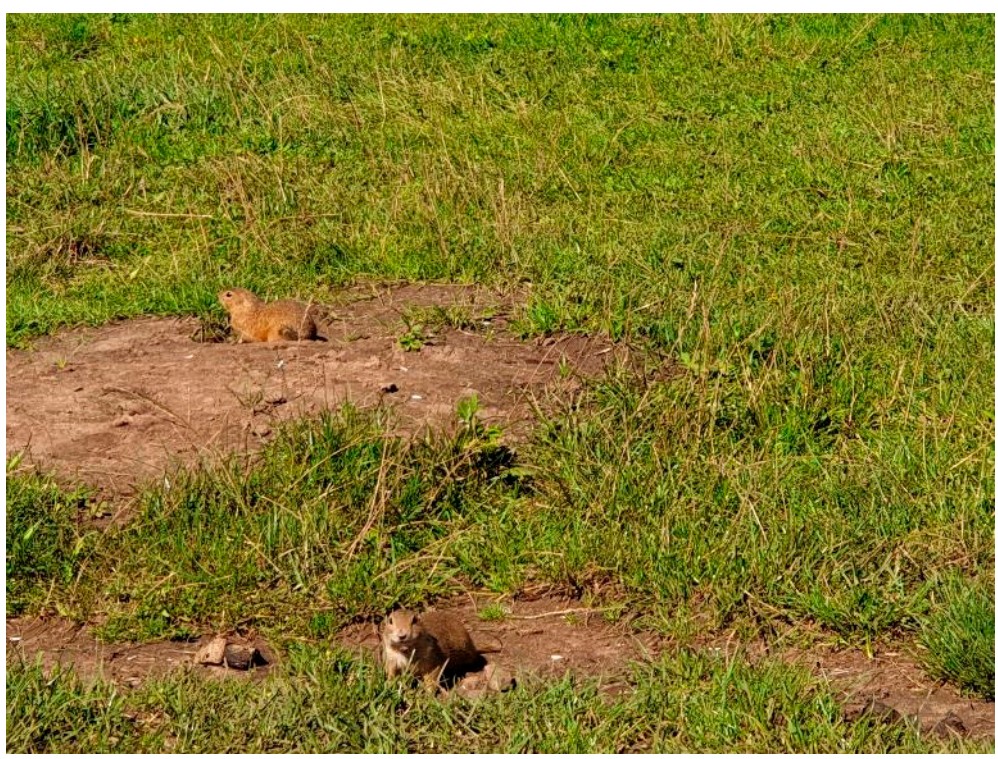

Ground squirrel in front of the burrow.

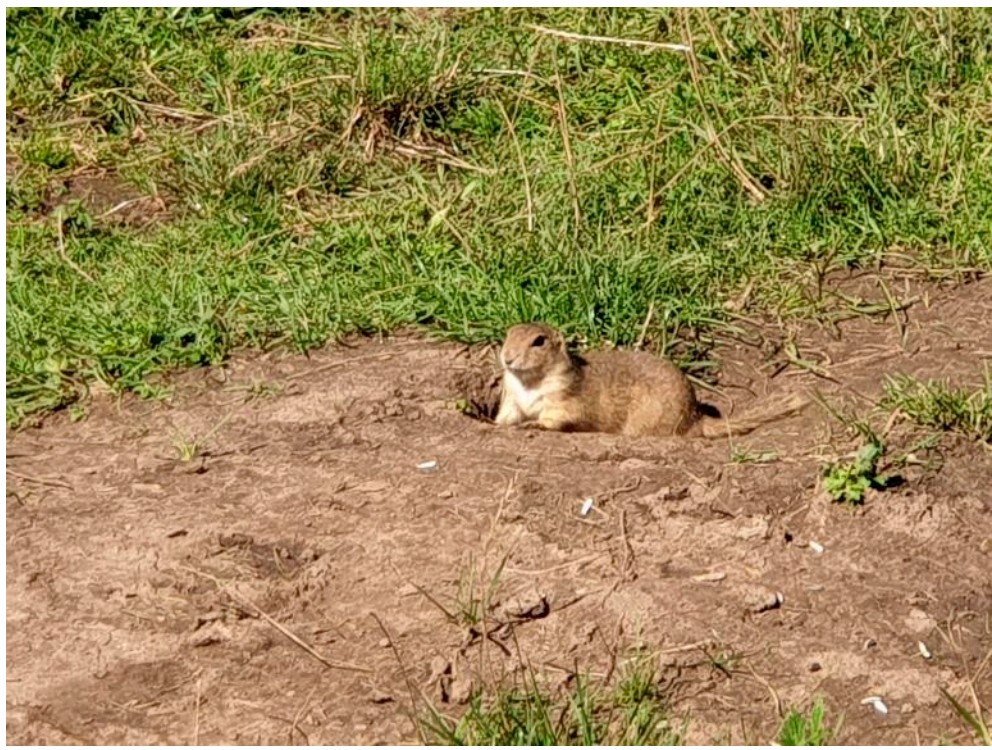

Ground squirrel running out of a burrow.

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
