# Peer review of "An Anthropogenically Created Landscape as a Habitat for the European Ground Squirrel Population Using the Example of the Muránska Planina National Park in the Western Carpathians (Slovakia)"

_land, doi:10.3390/land12112070_

Round 1

Reviewer 1 Report (Previous Reviewer 2)

Comments and Suggestions for Authors

I have reviewed the manuscript titled " Anthropogenically created landscape as a habitat for the European Ground Squirrel population on the example of the Muránska Planina National Park in the Western Carpathians (Slovakia)". focuses on the historical-geographical research and reconstruction of an anthropogenically created cultural landscape in the Muránska Planina National Park. The study aims to examine the potential of this landscape as a habitat for the European ground squirrel (Spermophilus citellus).

I have the following observations on this MS.

The MS does not contribute new in terms of methodology - a set of well-known methods have available for change scenarios in terms of land and deriving forces and these methods are important as well.

The introduction is weak, and the method section is trivial and vague in places. More recent literature work is required.

I fail to see a fruitful discussion on the generated datasets. But the introduction must be improved and the scientific problem has to be clearly identified and addressed.

I do not see little novelty in both scientific findings or methodological improvement.

"I have observed that the author has submitted an edited version of the manuscript in response to the reviewer's previous comments, which mainly addressed spelling mistakes and wording. However, I share a significant concern regarding the manuscript's scientific contribution, which remains unclear. The study lacks novelty and fails to introduce any new insights or advancements in the field. Based on these grounds, I am regrettably rejecting this manuscript."

Author Response

Reviewer 2 Report (New Reviewer)

Comments and Suggestions for Authors

Comments and suggestions for the authors

The study is interesting. However, it is not very clear to me why this long historical-biogeographical reconstruction of the landscape is needed to explain the ground squirrel population dynamics. I suggest specifying clearly the research gap.

Some general suggestions are:

- Write the subtitles of methodology with more appropriate names (lines 133 and 164)

- Delete information from lines 137 to 153 or show them in a table. The authors might consider using the methodology examples in the Discussion section.

- Delete the title on line 218

- In the results section, a timeline highlighting the main events would help to guide the reader.

- Write Fagus sylvatica correctly (line 307)

- Figures 3, 4, 7, and 8 can be presented as Figure 12 to compare the land use change.

- Separate caption Figure 9,10 (line 417-419)

Author Response

Reviewer 3 Report (New Reviewer)

Comments and Suggestions for Authors

This paper analyses the dynamics of land cover change from the 14th century to the present day at a site in the Western Carpathians (Slovakia). It relates these changes to the presence of habitat suitability for the European ground squirrel (Spermophilus citellus). I think the approach of the article is not very successful, although the idea of the MS is very interesting.

Throughout the text (even after correcting previous reviews) there are major problems with understanding what the various texts are referring to, text flow, and extended text on topics that are often difficult to understand what they are referring to if you didn't read 20-30 lines to connect them to the previous one. Without being copy-paste, too much text is essentially repeated as information in many parts of the text (regardless of the relevant section).

A more serious problem is the extremely small study area (essentially just one small site in a mountainous landscape with many similar valleys interspersed between forested slopes) that is used to make broad generalizations about the temporal influence of humans and local climate on shaping the landscape and the presence of a small mammal.

To be honest, I didn't quite understand the attempt to link such a historical search for landscape changes to the historical spread of this species (especially when it would have been impossible to have such information in the 14th and 15th centuries). Back then, on a much larger scale, with more natural space available for wildlife, the dynamics of species distribution of animals and spatial change of habitat were different.

This species has declined sharply throughout its range in recent decades, and many countries are attempting to reintroduce populations in previously extirpated areas by translocating individuals. The authors would then be able to model the optimal environment for the species in the area and correlate this (with far less textual detail) with the temporal presence of these landscapes in the area, in order to prioritize conservation sites.

I have made a number of comments so that MS can be further enhanced. You can find all my comments in the attachment. In italics

Round 2

Reviewer 1 Report (Previous Reviewer 2)

Comments and Suggestions for Authors

Significant changes have been made from the original submitted version, including additions to the introduction, methods, results, and discussion sections. These substantial revisions mean the manuscript needs to be re-reviewed in its current form.

In particular, the authors have added information to address previous concerns about novelty and contribution. However, the new additions require careful examination to determine if they satisfactorily establish the significance of this work.

Furthermore, the discussion and conclusions have been expanded and refocused, changing the overall implications of the study. Therefore, the revised manuscript needs assessment to ensure the revisions are logical and supported by the results presented.

Here are some suggestions to address the reviewer's concerns about novelty and contribution:

1.      Clearly highlight the knowledge gap your study is addressing in the introduction. What new insights about the creation and development of ground squirrel habitat does your historical analysis provide? How does reconstructing the land use history fill an important gap? Please follow the literature review by a clear and concise state of the art analysis. Clearly discuss what the previous studies that you are referring to. Recent research works need to be cited. There are few manuscripts suggested for this research:

https://doi.org/10.1080/19475683.2023.2166989

https://doi.org/10.3390/rs12030357

2.      Emphasize the significance of using a detailed historical lens to understand habitat change over time. This provides a nuanced understanding compared to only looking at recent landscape changes.

3.      Consider framing your study as a model or case study for using historical methods to reconstruct suitable habitats for other endangered species. The methods could be applied to other contexts.

4.      Discuss how your findings can inform ground squirrel conservation and management in the region going forward. What implications does this historical understanding have for current or future conservation efforts?

5.      Elaborate on the discussion to bring out new insights from your results. Don't just restate the findings - reflect on what is learned about long-term habitat change and stability, effects of past human activities, and parallels or differences to the present.

6.      Highlight unique aspects of the study area or findings that differentiate it from previous works. For example, focus on the Western Carpathians as an under-studied region.

To summarize, I recommend major revision and invite resubmission of a manuscript that has been thoroughly edited and peer reviewed in its current revised state.

Author Response

This manuscript is a resubmission of an earlier submission. The following is a list of the peer review reports and author responses from that submission.

Round 1

Reviewer 1 Report

Comments and Suggestions for Authors

Dear Editor, Dear Authors,

The topic of the manuscript (“Anthropogenically created landscape as a habitat for the European Ground Squirrel population on the example of the Muránska planina national park in the Western Carpathians (Slovakia)”) might be of interest to the Land journal readership. However, several serious issues prevent its publication. There a two major concerns:

1)      Scientific orientation

2)      Research design and goals.

Ad 1) I fear the manuscript has not enough research orientation and makes only marginal contributions to knowledge and theory. The main concern is the small size of the study area which makes it difficult to get solid conclusions – I am no sure if such a small study area is a good sample of the landscape dynamics of the region.

Ad 2) First, the authors should set the aims of the study in a more clear way – I believe the results are no in line with the initial objectives of the research (presented in the introduction section). On the other hand, the structure of the manuscript needs to be improved; for example, a better description of the methodological procedure is required. There is also a lack of order in the ideas presented all over the manuscript.

Please, find all my comments and suggestions in the attachment.  

Author Response

Dear reviewer,

On behalf of my co-authors I am resubmitting the manuscript (with the ID land-2284399) entitled ‘ANTHROPOGENICALLY CREATED LANDSCAPE AS A HABITAT FOR THE EUROPEAN GROUND SQUIRREL POPULATION ON THE EXAMPLE OF THE MU-RÁNSKA PLANINA NATIONAL PARK IN THE WESTERN CARPATHIANS (SLOVAKIA)’. We appreciate the interest that you have taken in our manuscript and the constructive criticism you have given. Based on your evaluation, I am resubmitting our manuscript after extensive revisions. We have used for editing the paper track changes function.

Sincerely

B. Gregorová

Reviewer 2 Report

Comments and Suggestions for Authors

I have reviewed the manuscript “Anthropogenically created landscape as a habitat for the European Ground Squirrel population on the example of the Muránska planina national park in the Western Carpathians (Slovakia)”. I read the manuscript carefully, I think the manuscript still needs a lot of improvement before it can be published.

1. I suggest that authors refine the expression abstract, and highlight the innovation and necessity of research. Please add a sentence which shows the necessity of this study.

2. In the introduction, the length is too short and the description of the research background is too simplistic, which makes it difficult to understand the core logic that the author is trying to express and the hierarchy is not clear.

3. I do not think the authors make it very clear of their contributes to this field. Please follow the literature review by a clear and concise state of the art analysis. Clearly discuss what the previous studies that you are referring to. In addition, in the introduction, what are the Research Gaps/Contributions? Please note that the paper could be clearly research gap and novelty of the study.

4. The author needs to further explain the uniqueness of Anthropogenically created landscape as a habitat and the necessity of studying the Spatial-temporal long-term monitoring in this area. Your research methods are not clear enough.

5. In the results section, you should focus on the main points of your research rather than putting in all your images. In addition, I recommend authors to present your results in a different way, thus enhancing the manuscript.

6. Another obvious problem with this paper is lack of sufficient explanation of the simulation results. You need to explain your simulation results in detail and why you got such results.

7. In the discussion section, I recommend author adding several reasoning and comparison of the study finding with other similar published work through available in the literature. In addition, please link your empirical results with a broader and deeper literature review.

8. The conclusions should be expanded as it doesn’t refer to the finding. The conclusion needs to add some specific values. In the conclusions, except to summarizing the actions, please strengthen the explanation of their significance, especially those stemming from previous work to make the findings and contributions of the paper clearer.

Author Response

(The authors gave the same response as above.)

Reviewer 3 Report

Comments and Suggestions for Authors

Dear Authors

Unfortunately I need to reject your article because I have got a lot of serious mistakes. I suggest to improve the paper and try again. Good luck and don't give up.

My remarks I will explain line by line:

2-3 it should be written: European ground squirrel instead of European Ground Squirrel

3-4 it should be written: Muránska Planina National Park instead of Muránska planina national park (the same like in 14 & 15 lines)

23 - European ground squirrel and elsewhere in the text (look: here in 26 line is ok)

31 and elsewhere - it should be written Biele Vody instead of Biele vody. Am I right?

49, 79, etc. - all Latin names in Italics: Spermophilus citellus

50 - insert European...

58-59 - instead of [3, 4, 5, 6, 7] it should be written [3-7]. It is a basic requirement for citing literature in MDPI. You should be more familiar with it. The same in 145-146 and elsewhere.

61-62 - it should be: .... 2,500 m a.s.l. [3, 8].

67 - [1, 9]

107, 108: two sentences, one by one, have the same begininng: "It is located in...". It doesn't look well.

Figure 1 - yellow colours should not cover countries borders. Instead of blue arrow (it is rather violet) you should put a circle.

3.1. too long chapter. We cite literature not to describe some things in details, so e.g. The selection of the square shape of the model territory is already a proven methodology, which was used by, e.g. J. Kolejka (2002) [13] or even: by [13]. (deleting: during the study of landscape changes of the Nové Mlýny water reservoir in the Czech Republic).

Do the same in others sentences in these paragraphs. It is too long and too boring.

3.2. 158-165 - please delete this paragraph. When you publish a paper in scientific journal these things are understandable for readers... respect readers time...

170-171 - repetition of 'crtitical content', English should be improve strongly.

185-187 - like above, leave only numbers in [ ] 

4:  207-227: I don't understand the idea of this chapter. After Methodology and before results.... Maybe you should remove these information to introduction or discussion? 

273 - [72, 73, 74, 75, 76, 77, 78] = [72-78]

331-348 - are these your research? I'm not sure... It is not discussion yet.

359-365 - you used four times "we can". 

378-388 - these information are really needed here, in Results chapter? Rather not.

446 - Figure two - delete this figure, because you put it also later.

Remarks to figure 3, 4, 7 and 8 arrange these figures in the same style like in figure 13: put together maps,  insert just one scale bar, and legend (now you even did not use the same colours in all 4 or font .... in legends). 

scale bar - 0-1 km is enough, not so detailed and 0-1,5 km.

507, 521, 535 - (Figure...) not (figure...)

541 - what source? 

Figure 9 and 10 -put stronger lines, especially in fig. 10.

549 - with visible ground squirrel burrows - try to make them on map more visible

Figure 11 should be localized in the middle, not to the left

561 - in whole paper is a mess, e.g. here: "Based on our research conclusions,..." Here we are in Results right?

617 - in figure 1 you explained colours, so you shoud do the same here.

633 - you did not any computer modelling, just put  land cover types.

674-675- our research, we can... English of the paper is poor.

Figure 14 - unfortunatelly, but in each map the research area (violet) is in different place of square.... it doesn't have sense....

723 - rather Museum....

828-833 - probably you forgot to delete it.

References should be prepared more precisely.

In general the paper in that form is too long and too boring. After acceptance the article to make a reviewer I was curious and full of hopes... Now I'm dissapointed.

Author Response

(The authors gave the same response as above.)
